# Why Agentic Theorem Prover Works:
# A Statistical Provability Theory of Mathematical Reasoning Models

**Sho Sonoda** [1 2]  **Shunta Akiyama** [1]  **Yuya Uezato** [1 3]

## Abstract

Agentic theorem provers combine a reasoning model, retrieval, search, and a proof assistant verifier, yet it remains unclear which components actually improve finite-budget proof success and why they help on real mathematical workloads. We study this question through *statistical provability*: the probability of reaching a verified proof within a budget on a specified stream of theorem instances. We model formal proof search as a finite-horizon reachability MDP with deterministic verifier dynamics, and show that under a faithful state abstraction the optimal success probability coincides with ordinary syntactic provability. We then analyze a simple but practically important pipeline: depth-wise offline action-value regression followed by greedy test-time proving. Our main theorem bounds the provability gap between the learned prover and the optimal prover by an occupancy-weighted sum of uniform action-value errors; in the common uniform-error reading, the leading complexity multiplier is the learned prover's average truncated proof length. The error decomposes into approximation error, geometric coverage of the training distribution, and Monte Carlo label noise, and improves to a fast rate under an action-gap margin condition. The result gives a component-sensitive account of why verifier feedback, retrieval, representation geometry, and proof-shortening mechanisms help on biased theorem workloads, without contradicting classical worst-case hardness.

## 1. Introduction

Large language models have improved rapidly at mathematical reasoning, and recent systems increasingly translate that capability into *verified* proofs by combining an LLM policy with retrieval, search, and a proof assistant verifier (Polu & Sutskever, 2020; Zheng et al., 2022; Yang et al., 2023; Jiang et al., 2023; Xin et al., 2024; Ren et al., 2025; Baba et al., 2025; Chen et al., 2025; Varambally et al., 2025; Achim et al., 2025). These agentic theorem provers are empirically effective, but the usual theoretical vocabulary does not explain their behavior well. Classical logic and proof theory ask whether a proof exists. Complexity theory asks how hard proof search can be in the worst case. A deployed prover faces a different question: *how likely is this particular algorithm to generate a verifier-accepted proof within a fixed compute budget on the problems it actually receives?*

This distinction matters. In standard proof theory, provability means the existence of a finite proof diagram in a formal calculus: for assumptions $\Gamma$, a conclusion $\varphi$, and a proof system $K$, the judgement $\Gamma \vdash_K \varphi$ is true if such a diagram exists. That is the right logical notion, and we do not replace it. But it is not enough to explain why an LLM-guided prover works. A proof may exist while a finite-budget search procedure almost never finds it; conversely, a learned prover may succeed often on the theorem families it is trained for even though unrelated worst-case instances remain hard.

The mechanism we study is simple. Real theorem-proving workloads are not uniform samples from all formal statements of a given size. They are biased toward library theorems, benchmark families, reused definitions, and recurring proof patterns. Training traces and verifier interaction expose this bias: they show which actions tend to make progress on the states that actually occur. If a learned scorer estimates those local success probabilities accurately on the high-mass region visited by the prover, greedy or search-based proving can have high finite-budget success probability without contradicting undecidability or worst-case lower bounds.

We call this algorithmic success probability *statistical provability*. For a problem stream $q_0$, a verifier-call budget $B$, and a prover policy $\pi$, it is the probability that the prover reaches a verified proof within $B$ calls, averaged over $X_0 \sim q_0$ and over the prover's own randomness. This

[1]CyberAgent, Inc. [2]RIKEN [3]National Institute of Informatics, Japan. Correspondence to: Sho Sonoda <sho.sonoda@riken.jp>.

*Proceedings of the 43rd International Conference on Machine Learning*, Seoul, South Korea. PMLR 306, 2026. Copyright 2026 by the author(s).

quantity depends on both the problem bias and the algorithm. It is therefore the object one needs in order to ask why an agentic prover works on its intended workload. Alongside success probability, we track the prover's average truncated proof length: the expected number of verifier calls made before success or budget exhaustion. This is directly readable from proof traces and is the main interpretability handle in the bound: for a fixed local value-estimation error, shorter average proofs imply smaller end-to-end loss.

Our analysis formalizes this question with a finite-horizon reachability MDP. The state is the current proof obligation together with verifier feedback, the action is the next prover step, and the transition is the verifier's deterministic response. The Bellman structure of this MDP turns successful proving into an action-value estimation problem. We analyze a simple pipeline aligned with current practice: learn depth-wise action-value functions offline from verifier-backed rollouts, then prove greedily at test time using the learned scores. The resulting theorem shows how approximation error, representation geometry, data coverage, rollout budget, and action-gap margins affect end-to-end proof success.

**Contributions.**

- We formalize agentic theorem proving as a finite-horizon reachability MDP and define *statistical provability* as the finite-budget probability that a specified prover generates a verified proof on $X_0 \sim q_0$.

- We show that, under a faithful abstraction of proof states, optimal MDP success is exactly syntactic provability within the same verifier budget. This keeps the logic intact while enabling value-based analysis.

- We analyze *offline action-value regression + greedy proving* and derive a provability bound whose leading multiplier is the learned prover's average truncated proof length and whose statistical term separates approximation error, coverage geometry, and Monte Carlo label noise.

- We interpret the theorem in terms of concrete design choices: retrieval, verifier feedback, representation learning, and proof-shortening mechanisms help by reducing average proof length, improving margins, or localizing the learning problem to better covered regions.

## 2. Proof Search as a Reachability MDP

**Ordinary proof systems.** Fix a formal proof system $K$. At the logical level, one can think of $K$ as specifying a language of formulas, a set of axioms, and a set of inference rules. A $K$-proof of a conclusion $\varphi$ from assumptions $\Gamma$ is a finite proof tree whose leaves are axioms or assumptions, whose internal nodes follow the inference rules, and whose root is $\varphi$. When such a proof tree exists, we write $\Gamma \vdash_K \varphi$. This is the standard proof-theoretic notion of *provability* used in logic.

**Proof assistants and proof-search states.** A proof assistant implements such a formal system together with a trusted verifier. Lean (de Moura & Ullrich, 2021) is a representative example: its kernel checks proof objects, while users and agents usually interact through tactics. During backward proof search, the visible state is a finite list of open goals, each of the form "under local context $\Gamma_i$, prove $\varphi_i$." A tactic transforms this list into a new list of goals, and the proof is complete when the list is empty. Thus a proof assistant already exposes the interface needed for a sequential decision problem: current proof state, proposed action, verifier-approved next state. This is not specific to backward search. In forward proof search, the state can instead be the set of formulas derived so far together with the target, an action selects an inference-rule instance and its premises, and the verifier transition adds the derived conclusion. Figure 1 illustrates these two views on the same modus-ponens proof and labels the corresponding MDP components.

**Verifier interface.** We abstract the interface in Figure 1 as follows. Let $\mathcal{X}$ be the proof-state representation. A state contains the currently open obligations together with local context, metavariables, and verifier messages. Let $\mathcal{A}$ be an action space of prover moves such as tactics, inference-rule instances, lemma invocations, or retrieval-conditioned commands. We assume a deterministic verifier update

$$F : \mathcal{X} \times \mathcal{A} \to \mathcal{X},$$

where invalid actions are mapped to explicit failure states. In forward natural deduction, for example, the action "apply modus ponens to $A \to B$ and $A$" maps a state containing those facts to a state that also contains $B$. In Lean tactic mode, tactics construct proof terms incrementally: `apply` unifies the conclusion of a supplied expression with the current goal and creates subgoals for missing arguments, while `exact` fills the current goal exactly. Thus a tactic command is naturally an action whose verifier-approved result is a new proof state. The same abstraction also covers backward sequent search, Rocq, and Isabelle.

For a budget $B \in \mathbb{N}$, let $\mathrm{Prov}_B(x) \in \{0, 1\}$ denote whether a closed proof can be reached from $x$ within at most $B$ verifier calls. This is the budgeted analogue of ordinary proof-theoretic provability. It is still a pointwise logical property of the initial state and the verifier, not a statement about whether a learned prover will find the proof.

**Proposition 2.1** (Verifier interfaces induce deterministic MDPs). *Any proof-search system specified by a state space*

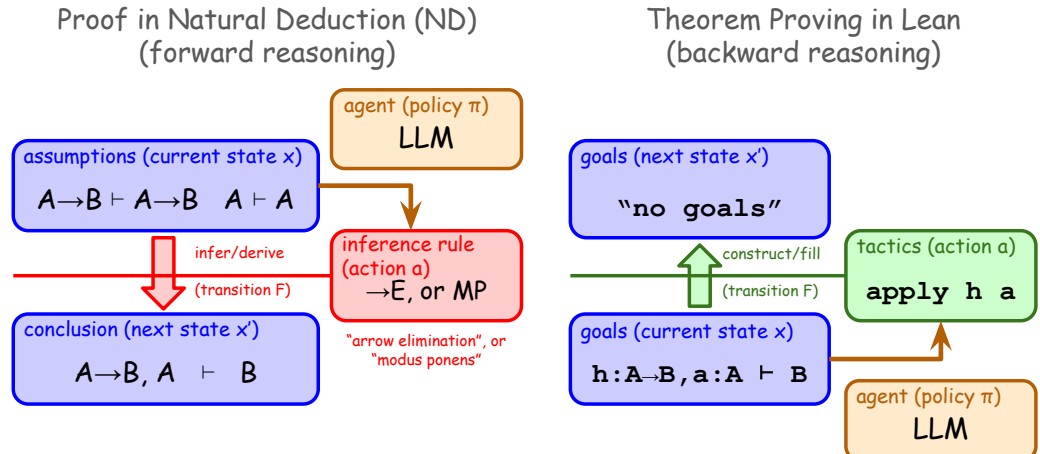

**Figure 1.** Formal proving as a verifier-defined MDP. Left: in forward natural-deduction search, the current state $x$ records available assumptions or derived facts; an action $a$ selects an inference-rule instance such as implication elimination (modus ponens); and the transition $F(x, a) = x'$ adds the derived conclusion, here $B$. Right: in Lean-style backward search, the current state $x$ is the open goal $h : A \to B, \ a : A \vdash B$; the tactic action `apply h a` supplies an exact proof term for that goal; and the verifier transition returns the solved state $x'$ with no remaining goals. The diagram suppresses many implementation details, but the MDP correspondence is always state, action, verifier transition, solved set.

$\mathcal{X}$, an action space $\mathcal{A}$, a verifier update map $F : \mathcal{X} \times \mathcal{A} \to \mathcal{X}$, and a solved set $\mathsf{G} \subseteq \mathcal{X}$ induces a deterministic finite-horizon reachability MDP via

$$P(\cdot \mid x, a) = \delta_{F(x,a)}.$$

This proposition is mainly a bookkeeping device. The MDP is not a new logic; it is the verifier's step-by-step operational semantics written in the notation needed for dynamic programming. Randomness may enter through stochastic policies, exploration, retrieval, or decoding, but the verifier transition itself is deterministic in the systems we target.

**Why whole proof states, not single goals?** One proof step can create several subgoals that must all be discharged. The induced search object is therefore closer to an AND-OR graph or directed hypergraph than to a single trajectory. Modeling the state as the full multiset of currently open goals restores the Markov property: all future obligations are summarized by the current proof state, and branching is handled inside the verifier update.

**State representations and statistical provability.** A convenient proof-state representation is a finite measure over embedded open goals, but the main theorem only needs a compact metric representation on the region where training and testing concentrate. For a policy $\pi$, define the hitting time

$$T_{\mathsf{G}} := \inf\{t \geq 0 : X_t \in \mathsf{G}\}, \qquad X_{t+1} = F(X_t, A_t).$$

The budgeted success probability is

$$V_B^{\pi}(x) := \mathbb{P}_x^{\pi}(T_{\mathsf{G}} \leq B), \qquad V_B^*(x) := \sup_{\pi} V_B^{\pi}(x).$$

Given an initial-state distribution $q_0 \in \mathcal{P}(\mathcal{X})$, we define

$$\mathrm{SP}_B^{\pi}(q_0) := \mathbb{E}_{q_0}\left[V_B^{\pi}(X)\right], \quad \mathrm{SP}_B^*(q_0) := \mathbb{E}_{q_0}\left[V_B^*(X)\right].$$

We call these quantities *statistical provability*. The companion length statistic is the average truncated proof length

$$\mathrm{Len}_B^{\pi}(q_0) := \mathbb{E}_{q_0}^{\pi}[T_{\mathsf{G}} \wedge B] = \sum_{s=1}^{B} \mathbb{P}_{q_0}^{\pi}(T_{\mathsf{G}} \geq s). \tag{1}$$

where the equality is the tail-sum formula. Operationally, $\mathrm{Len}_B^{\pi}(q_0)$ is the average number of verifier calls spent by prover $\pi$ before it either finds a proof or exhausts the budget. Unlike worst-case proof length, it is estimated from the same traces used to evaluate a prover, which makes it a useful quantity for interpreting the theorem.

**Learning setup and notation.** We use roman letters for data-generating objects and Greek letters for learned policies. The initial theorem distribution is $q_0$. At remaining horizon $t$, the training query pairs are drawn from a data-query distribution $q_t$ supported on a relevant compact state-action region $\mathcal{C}_t$. The target of regression is the optimal one-step value

$$Q_t^*(x, a) := V_{t-1}^*(F(x, a)).$$

The learning side consists of a hypothesis class $\mathcal{H}_t$, an estimator $\widehat{Q}_t \in \mathcal{H}_t$, and the induced greedy policy

$$\hat{\pi}_t(x) \in \arg\max_{a : (x,a) \in \mathcal{C}_t} \widehat{Q}_t(x, a).$$

We write $T_{\mathsf{G}}$ for the hitting time of the solved set and $O_s^{\hat{\pi}}$ for the conditional occupancy law given non-solution. The full statistical assumptions on $q_t, \mathcal{H}_t, \widehat{Q}_t$, and the rollout labels are stated in Assumption 3.1.

**Proposition 2.2** (Faithful abstraction preserves provability). *Suppose a lower-level state space $\mathcal{R}$ with verifier update $F_{\mathcal{R}} : \mathcal{R} \times \mathcal{A} \to \mathcal{R}$ and solved set $\mathsf{G}_{\mathcal{R}}$ is represented in $\mathcal{X}$ by an encoding map $e : \mathcal{R} \to \mathcal{X}$. Assume that the encoding is faithful in the sense that*

$$e(F_{\mathcal{R}}(r,a)) = F(e(r), a)$$

*for every state-action pair in the lower-level representation, and that solved states are preserved:*

$$r \in \mathsf{G}_{\mathcal{R}} \iff e(r) \in \mathsf{G}.$$

*Let $\mathrm{Prov}_B(r)$ denote reachability of $\mathsf{G}_{\mathcal{R}}$ from $r$ within $B$ lower-level verifier calls. Then for every $r \in \mathcal{R}$,*

$$V_B^*(e(r)) = \mathbf{1}[\mathrm{Prov}_B(r) = 1].$$

*Consequently, if $q_0$ is the push-forward of a problem distribution on $\mathcal{R}$, then $\mathrm{SP}_B^*(q_0)$ is exactly the mass of instances that are syntactically provable within budget $B$.*

**Why this matters.** Proposition 2.2 says the MDP language does not replace the underlying logic; it repackages it in a form compatible with dynamic programming and statistical learning. This is the bridge between classical provability and the finite-budget generation probability that matters for agentic systems.

**Encodings and concentration regions.** The theorem below only needs a metric representation of the proof states on the state-action region where training and test-time comparisons take place. One possible implementation is a measure-valued encoding of the current multiset of open goals, but the analysis is stated directly in terms of the concentration sets $\mathcal{C}_t$. Details about variable numbers of goals and high-probability truncation are deferred to Appendix B.

## 3. Offline Action-Value Learning

We now analyze a depth-wise value-learning pipeline. For remaining horizon $t = 1, \ldots, B$, define the optimal one-step action-value function

$$Q_t^*(x,a) := V_{t-1}^*(F(x,a)).$$

Given estimators $\widehat{Q}_t$, the learned prover acts greedily:

$$\hat{\pi}_t(x) \in \underset{a:(x,a)\in\mathcal{C}_t}{\arg\max} \widehat{Q}_t(x,a).$$

This deliberately isolates the central scoring problem. Retrieval, decomposition, reranking, and tool feedback matter in our theory only through how much they improve these action-value estimates or shorten the remaining proof.

**Why action-value learning rather than policy learning?** The theorem-proving pipelines that motivate this paper are usually not trained end to end as policies over full proof trees. In practice, many components act more like *scorers* or *rerankers*: given a state and a candidate action, they estimate whether this move is likely to lead to a proof under the remaining budget. Our formulation matches that reality. Training is offline regression against verifier-backed success targets, while test-time proving is greedy decoding with respect to the learned action values.

**Assumption 3.1** (Depth-wise offline value-learning model). Fix a budget $B$ and an initial-state distribution $q_0$. For each remaining horizon $t = 1, \ldots, B$, assume:

1. There is a compact concentration set $\mathcal{C}_t \subseteq \mathcal{X} \times \mathcal{A}$ containing both the training query pairs and the state-action comparisons performed by the learned prover. For every relevant state $x$ at which the regret bound is evaluated, the slice $\mathcal{A}_t(x) := \{a : (x,a) \in \mathcal{C}_t\}$ is nonempty, the greedy action $\hat{\pi}_t(x) \in \arg\max_{a \in \mathcal{A}_t(x)} \widehat{Q}_t(x,a)$ exists, and there is an optimal Bellman action $a_t^*(x) \in \mathcal{A}_t(x)$ such that

   $$Q_t^*(x, a_t^*(x)) = \sup_{a \in \mathcal{A}} Q_t^*(x,a).$$

2. $Q_t^*$ is $L_{Q,t}$-Lipschitz on $\mathcal{C}_t$, and the hypothesis class satisfies $\mathcal{H}_t \subseteq \mathrm{Lip}_{L_{H,t}}(\mathcal{C}_t, [0,1])$.

3. Query pairs $(X_{t,1}, A_{t,1}), \ldots, (X_{t,N_t}, A_{t,N_t})$ are i.i.d. samples from an exploration distribution $q_t$ supported on $\mathcal{C}_t$, and $q_t$ has local mass lower bound

   $$q_t(B(z,r)) \geq c_t r^{d_t}$$

   for all $z \in \mathcal{C}_t$ and sufficiently small $r > 0$, where $B(z,r)$ is the metric ball in $\mathcal{C}_t$. Here $N_t$ is the number of sampled state-action query pairs, $d_t$ is the effective local dimension, and $c_t > 0$ is the corresponding coverage constant.

4. Each query pair receives $m_t$ conditionally independent binary rollout labels $Y_{t,i,1}, \ldots, Y_{t,i,m_t} \in \{0,1\}$ such that

   $$\mathbb{E}[Y_{t,i,j} \mid X_{t,i}, A_{t,i}] = Q_t^*(X_{t,i}, A_{t,i}).$$

   Thus $m_t$ is the number of independent verifier-backed rollout labels collected for each sampled query pair.

5. Writing $\bar{Y}_{t,i} := m_t^{-1} \sum_{j=1}^{m_t} Y_{t,i,j}$, the estimator is chosen by minimax regression on the sampled points:

   $$\widehat{Q}_t \in \underset{h \in \mathcal{H}_t}{\arg\min} \max_{1 \leq i \leq N_t} |h(X_{t,i}, A_{t,i}) - \bar{Y}_{t,i}|.$$

**Relevant domains and stability.** The set $\mathcal{C}_t$ should be read as the non-negligible state-action region occupied by the data generator and by the learned prover at remaining horizon $t$. Empirically, it can be approximated by the support, or a high-mass enlargement, of the available prover traces. The assumption is not automatically stable under arbitrary model changes: if a new prover leaves this occupied region, the bound no longer certifies its behavior without additional coverage assumptions. This is why the theorem is a trace-local diagnostic rather than a global claim about the entire proof-search space.

**Optimizer and retention conditions.** The optimizer clauses in Assumption 3.1 have different roles. The existence of the greedy action is a mild well-posedness condition: once $\mathcal{A}_t(x)$ is nonempty, compactness of the slice and Lipschitz continuity of $\widehat{Q}_t$ give a maximizer by Weierstrass' theorem. The optimal Bellman-action clause is stronger. It says not only that an optimal first action exists, but also that at least one such action lies inside the candidate region $\mathcal{C}_t$ on the states where the learned prover is evaluated. This is a candidate-retention assumption: the data generator and retrieval mechanism must not exclude every optimal move from the local comparison set. Standard compact/Feller conditions that guarantee Bellman maximizers are recalled in Section A.3; the main theorem only needs their trace-local consequence stated in Assumption 3.1. If one has only an $\eta_t$-optimal retained action, the same proof gives the bound with an additional occupancy-weighted $\eta_t$ error term.

**Regularity and optimization idealizations.** The Lipschitz assumptions on $Q_t^*$ and $\mathcal{H}_t$ are representation assumptions: they assert that, on the occupied state-action region, the chosen encoding makes nearby proof states have comparable future success probabilities and makes the learned score class no rougher than $L_{H,t}$. The exact minimax regression step is also an idealization. Allowing a numerical or statistical optimizer whose empirical minimax loss is within $\xi_t$ of the infimum simply adds a corresponding $\xi_t$ term to the value-estimation error $\varepsilon_t$.

**Rollout targets.** The binary labels $Y_{t,i,j}$ are idealized verifier-backed success labels: after proposing action $A_{t,i}$ at state $X_{t,i}$, one continues for the remaining budget and records whether a verified proof is reached. The assumption $\mathbb{E}[Y_{t,i,j} \mid X_{t,i}, A_{t,i}] = Q_t^*(X_{t,i}, A_{t,i})$ is an oracle version of this construction. In practice, repeated rollouts, teacher-guided rollouts, or bootstrapped value targets only approximate this oracle target; the resulting bias is absorbed into the approximation term in $\varepsilon_t$.

**Theorem 3.1** (Offline action-value regression implies statistical provability). *Under Assumption 3.1, for any $\delta_1, \ldots, \delta_B \in (0,1)$, with probability at least $1 - \sum_{t=1}^{B} \delta_t$*

*over the sampled query pairs and rollout labels,*

$$\mathrm{SP}_B^{\hat{\pi}}(q_0) \geq \mathrm{SP}_B^*(q_0) - 2\sum_{s=1}^{B} \mathbb{P}_{q_0}^{\hat{\pi}}(T_{\mathsf{G}} \geq s)\, \varepsilon_{B-s+1}, \quad (2)$$

*where the probabilities $\mathbb{P}_{q_0}^{\hat{\pi}}(T_{\mathsf{G}} \geq s)$ form the learned prover's unsolved-mass curve, and where*

$$\varepsilon_t := \underbrace{\inf_{h \in \mathcal{H}_t} \|h - Q_t^*\|_{L_\infty(\mathcal{C}_t)}}_{approximation}$$

$$+ \underbrace{C_{\mathrm{cov}}(L_{H,t} + L_{Q,t})c_t^{-1/d_t}\left(\frac{\log(4N_t/\delta_t)}{N_t}\right)^{1/d_t}}_{coverage\,/\,geometry}$$

$$+ \underbrace{2\sqrt{\frac{\log(4N_t/\delta_t)}{2m_t}}}_{rollout\ noise}. \quad (3)$$

*Here $C_{\mathrm{cov}} > 0$ is a universal covering constant. The three terms in $\varepsilon_t$ are, respectively, approximation error, finite-sample coverage error over $\mathcal{C}_t$, and rollout-label noise. If in addition the conditional occupancy distribution*

$$O_s^{\hat{\pi}} := \mathcal{L}(X_{s-1} \mid T_{\mathsf{G}} \geq s)$$

*satisfies the margin condition, for every $s = 1, \ldots, B$ and every $u \geq 0$,*

$$O_s^{\hat{\pi}}(\Delta_t \leq u) \leq C_\Delta u^\beta, \qquad t = B - s + 1,$$

*for some constants $C_\Delta > 0$ and $\beta \geq 0$, where $\Delta_t(x)$ is the gap between the best and second-best candidate action values at remaining horizon $t$, then on the same event,*

$$\mathrm{SP}_B^{\hat{\pi}}(q_0) \geq \mathrm{SP}_B^*(q_0) - 2C_\Delta \sum_{s=1}^{B} \mathbb{P}_{q_0}^{\hat{\pi}}(T_{\mathsf{G}} \geq s)$$
$$\cdot (2\varepsilon_{B-s+1})^{\beta+1}. \quad (4)$$

**Average-length reading.** The first bound is immediately interpretable because the probabilities $\mathbb{P}_{q_0}^{\hat{\pi}}(T_{\mathsf{G}} \geq s)$ sum to the learned prover's average truncated proof length in Equation (1). In particular, if $\sup_t \varepsilon_t \leq \bar{\varepsilon}_B$, then

$$\mathrm{SP}_B^*(q_0) - \mathrm{SP}_B^{\hat{\pi}}(q_0) \leq 2\bar{\varepsilon}_B \, \mathrm{Len}_B^{\hat{\pi}}(q_0).$$

Thus the theorem does not only say that smaller value-estimation error is better; it says where that error is accumulated. A prover that resolves typical instances quickly has a smaller multiplier, even if the local error level $\bar{\varepsilon}_B$ is unchanged.

**Proof ingredients.** The argument has three layers. First, a deterministic Bellman recursion expresses the optimal

finite-budget success probability in terms of the one-step action-value functions $Q_t^*$. Second, a uniform bound on $\widehat{Q}_t - Q_t^*$ yields a one-step greedy regret inequality, and telescoping this inequality along the learned trajectory produces the occupancy-sensitive bound in Equation (2). Third, the regression analysis shows that $\widehat{Q}_t$ is uniformly close to $Q_t^*$ on the relevant compact concentration set by combining sample coverage of the query pairs with concentration of the averaged rollout labels. Under a margin condition, only near-tie states can incur harmful misranking, which upgrades the linear dependence on $\varepsilon_t$ to the fast-rate form in Equation (4). Appendix A contains the full statements and proofs.

**Budget allocation.** Let $n_t := N_t m_t$ be the total verifier-label budget at depth $t$. Write

$$\varepsilon_{\mathrm{app},t} := \inf_{h \in \mathcal{H}_t} \|h - Q_t^*\|_{L_\infty(\mathcal{C}_t)}$$

for the approximation term in Equation (3). Balancing the geometry term and the rollout-noise term gives

$$N_t \asymp n_t^{d_t/(d_t+2)}, \qquad m_t \asymp n_t^{2/(d_t+2)},$$

which yields

$$\varepsilon_t = \varepsilon_{\mathrm{app},t} + \widetilde{O}\Big((L_{H,t} + L_{Q,t})c_t^{-1/d_t} n_t^{-1/(d_t+2)}\Big).$$

Thus the statistical rate depends on an *effective geometry* through $(d_t, c_t)$, not on the raw syntactic size of the logic.

**A shortest-proof interpretation.** Let $L^*(x)$ denote the shortest horizon $t$ such that $V_t^*(x) = 1$, with $L^*(x) = \infty$ if no budget-feasible proof exists. Then $\mathrm{SP}_B^*(q_0)$ is precisely the $q_0$-mass of instances with $L^*(x) \le B$. In other words, the target being learned is not arbitrary reward shaping; it is the probability mass of theorem instances whose shortest verified proofs fit inside the compute budget. This interpretation is useful when comparing proof-shortening mechanisms, because decomposition or lemma introduction can improve statistical provability either by increasing action-value regularity or simply by reducing the relevant shortest proof lengths.

**Corollary 3.1** (Interpretation on provable instances)**.** *If* $\sup_t \varepsilon_t \le \bar{\varepsilon}_B$, *then on the event of Theorem 3.1,*

$$\mathrm{SP}_B^{\hat{\pi}}(q_0) \ge \mathrm{SP}_B^*(q_0) - 2\bar{\varepsilon}_B \, \mathrm{Len}_B^{\hat{\pi}}(q_0).$$

*If moreover* $q_0(\mathrm{Prov}_B = 1) > 0$*, then by Proposition 2.2,*

$$\mathbb{E}_{q_0}\big[V_B^{\hat{\pi}}(X) \mid \mathrm{Prov}_B(X) = 1\big] \ge 1 - \frac{2\bar{\varepsilon}_B \, \mathrm{Len}_B^{\hat{\pi}}(q_0)}{q_0(\mathrm{Prov}_B = 1)}.$$

The corollary makes the comparison especially concrete: the theorem is not only about abstract value gaps. It lower bounds the learned prover's average success probability on the subset of instances that are syntactically provable within the budget.

## 4. What the Bound Says About Agentic Provers

**Average proof length matters through occupancy.** If $\sup_t \varepsilon_t \le \bar{\varepsilon}_B$, then Equation (2) implies

$$\mathrm{SP}_B^*(q_0) - \mathrm{SP}_B^{\hat{\pi}}(q_0) \le 2\bar{\varepsilon}_B \sum_{s=1}^{B} \mathbb{P}_{q_0}^{\hat{\pi}}(T_{\mathsf{G}} \ge s)$$

$$= 2\bar{\varepsilon}_B \, \mathrm{Len}_B^{\hat{\pi}}(q_0).$$

The regret therefore scales with $\mathrm{Len}_B^{\hat{\pi}}(q_0)$, the learned prover's average truncated proof length. Any mechanism that shortens proofs or resolves them earlier, such as useful decomposition or better subgoal selection, directly reduces the end-to-end loss multiplier.

It is worth stressing that the theorem depends on the *learned prover's* unsolved-mass curve, not directly on the distribution of optimal shortest proof lengths. This is unavoidable. A single early mistake can move the learned prover into a dead end or into a region where the remaining shortest proof is much longer than under an optimal policy. Without extra assumptions, the quantity that propagates through the Bellman argument is therefore the occupancy actually induced by the learned policy itself.

**Retrieval and representation help by reshaping the learning problem.** The geometry term in Equation (3) separates three effects: $L_{Q,t}$ captures how irregular the true action-value is under the chosen representation, $L_{H,t}$ captures how restrictive the score class is, and $c_t^{-1/d_t}$ captures how well the exploration distribution covers the region that matters. Retrieval, premise selection, and representation learning help when they make $Q_t^*$ smoother, concentrate the relevant state-action pairs on a lower dimensional region, or increase action margins by pruning obviously bad choices. This provides a concrete statistical interpretation of pipeline engineering choices that are often justified only empirically.

This also suggests a concrete representation-learning objective. A good state representation should not merely make theorem states easy to cluster; it should make the relevant action-value functions smoother and the training distribution thicker on the states that matter at test time. In that sense, the theorem gives a certificate-oriented criterion for deciding whether a representation is actually useful for proving.

**Bellman certificates.** The same Bellman inequalities can also be used to build upper and lower certificates for the optimal success probability $V_B^*$. This is not needed for the main theorem, whose role is to compare a learned prover with the optimal finite-budget prover, so we defer the certificate view and its representation-learning interpretation to Appendix C.

**Verifier feedback matters twice.** First, it supplies the rollout labels used for action-value regression. Second, at test time it prevents the prover from drifting silently into invalid branches. In our framework, better verifier feedback appears as smaller approximation error, better coverage of the relevant region, and larger action gaps. This explains why a strong verifier can improve scaling even when the base policy itself is unchanged.

**Variable numbers of goals do not break the theory.** Real proof states may generate many subgoals. The main theorem does not require global compactness of the entire state space; it only needs compact concentration sets $\mathcal{C}_t$ at each remaining horizon. A standard truncation argument adds an overflow penalty $\delta_W$ when one wants to reason on a globally compact state space. We state this extension in Appendix B.

**Why we focus on greedy proving.** Beam search, top-$k$ filtering, rollouts, and backtracking are important in practice. We focus on greedy execution because it exposes the clearest connection between value estimation and success probability. The same Bellman argument extends to richer planners once one adds candidate-retention or planner-suboptimality assumptions; we briefly discuss this route in Appendix D.

## 5. Easy and Hard Instance Regimes

The theorem organizes a common empirical observation: some theorem families become easy for agentic provers very quickly, while others remain stubbornly difficult even when the base reasoning model looks strong.

**Easier instances.** The favorable regime is characterized by small $\text{Len}_B^{\hat{\pi}}(q_0)$, large action gaps, low-dimensional concentration of reachable proof states, and good coverage by the exploration distribution. In that regime, $\varepsilon_t$ falls quickly with the verifier-label budget and the occupancy weights decay rapidly because many trajectories solve early. This is the setting where retrieval and decomposition are most useful: they can expose the right lemma early enough that the rest of the proof becomes almost deterministic.

**Harder instances.** The difficult regime is characterized by long proof horizons, many near-tie actions, poor training coverage, or proof states whose useful continuations depend on brittle symbolic details. Then both parts of the bound become unfavorable: the value-learning problem is statistically harder, and the occupancy weights stay large because errors made early can send the prover into dead ends. This is exactly where worst-case hardness and practical failure modes align.

Using $\bar{\varepsilon}_B$ for a uniform bound on the depth-wise errors, the main theorem can be read schematically as

$$\text{SP}_B^*(q_0) - \text{SP}_B^{\hat{\pi}}(q_0) \lesssim \text{Len}_B^{\hat{\pi}}(q_0)\, \bar{\varepsilon}_B$$

without a margin condition, and as

$$\text{SP}_B^*(q_0) - \text{SP}_B^{\hat{\pi}}(q_0) \lesssim \text{Len}_B^{\hat{\pi}}(q_0)\, \bar{\varepsilon}_B^{\beta+1}$$

under a margin condition. This captures the main engineering message using the same notation as the theorem: to improve proving success, one can reduce average proof length, reduce value-estimation error, or make correct actions more separated.

**Scaling-law viewpoint.** The theorem also yields immediate sample-complexity heuristics. Ignoring logarithmic factors and approximation error, the uniform regression bound behaves like

$$\varepsilon_t \approx n_t^{-1/(d_t+2)}.$$

Here $n_t = N_t m_t$ is the verifier-label budget at remaining horizon $t$, and $d_t$ is the effective local dimension from Assumption 3.1. If these quantities are roughly constant across horizons, say $n_t \approx n$ and $d_t \approx d$, then Equation (2) gives

$$\text{SP}_B^*(q_0) - \text{SP}_B^{\hat{\pi}}(q_0) \lesssim \text{Len}_B^{\hat{\pi}}(q_0)\, n^{-1/(d+2)}.$$

Under a margin condition with exponent $\beta$, the dependence improves schematically to

$$\text{SP}_B^*(q_0) - \text{SP}_B^{\hat{\pi}}(q_0) \lesssim \text{Len}_B^{\hat{\pi}}(q_0)\, n^{-(\beta+1)/(d+2)}.$$

These formulas clarify what kinds of apparent "test-time scaling" gains are actually plausible. Large practical improvements do not require changing worst-case complexity classes; they can arise whenever retrieval, decomposition, or representation learning reduce the effective horizon, lower the effective geometric dimension, or increase the action-gap margin.

This is also why the choice of $q_0$ matters. The same logical system may look intractable when $q_0$ is uniform or adversarial over hard instances, and much easier when $q_0$ concentrates on recurring definitions, library idioms, and proof patterns. Our framework does not deny the former; it tries to explain the latter.

## 6. Related Work

**From symbolic automation to learning-guided search.** Learning-guided theorem proving predates modern LLMs. TacticToe (Gauthier et al., 2017; 2021) learned tactic guidance and combined it with Monte Carlo tree search in HOL4. HOList and related systems (Bansal et al., 2019) established large-scale environments for higher-order theorem proving.

LeanDojo (Yang et al., 2023) lowered the barrier to training and evaluating Lean-based agents, and miniF2F (Zheng et al., 2022) helped standardize formal mathematics benchmarks. These works already support the sequential-decision view: proof search unfolds through adaptive interaction with a verifier under tight computational budgets. The MDP perspective is also explicit in Bourbaki (Zimmer et al., 2025), which studies self-generated, goal-conditioned MDPs and MCTS-like search for theorem proving. Our use of MDPs has a different purpose: the MDP is the mathematical interface through which we analyze finite-budget provability, Bellman certificates, and statistical score-estimation errors, rather than an implemented search framework.

**Neural language models and formal proving.** GPT-f (Polu & Sutskever, 2020) demonstrated early on that generative language models can discover useful formal proofs. The main methodological shift was from hand-engineered proof features to learned representations of states, tactics, and libraries. This made it natural to combine proposal, verification, and repair in multi-step loops rather than treating theorem proving as one-shot next-step classification.

**Agentic provers and decomposition.** Recent systems increasingly operate as explicit agentic workflows with proposal, retrieval, decomposition, verification, and refinement. DSP (Jiang et al., 2023) uses informal proof sketches to guide formal search. DeepSeek-Prover and DeepSeek-Prover-V2 (Xin et al., 2024; Ren et al., 2025) emphasize data generation and recursive subgoal decomposition in Lean 4. Prover Agent (Baba et al., 2025), Seed-Prover (Chen et al., 2025), Hilbert (Varambally et al., 2025), and Aristotle (Achim et al., 2025) all make intermediate lemmas or subgoals first-class objects. This decomposition trend also echoes classical proof complexity: useful cuts can dramatically shorten proofs, while their elimination can cause severe blow-up (Boolos, 1984). Our theory gives a statistical account of the same phenomenon through effective horizon and action-gap effects. This is complementary to Sonoda et al. (2026), which studies a more specialized flat versus hierarchical comparison and proves an exponential sample-complexity separation under cut-aware structure. The present paper instead develops a general value-based theory of time-bounded statistical provability and score-guided proving.

Seen from this angle, the current wave of agentic theorem proving is not just "LLMs applied to theorem proving." It is a convergence of three older ideas: proof search as sequential decision making, proof complexity as sensitivity to intermediate lemmas, and inference-time computation as a resource that can be adaptively allocated.

**Test-time scaling, verifiers, and interactive learning.** Our viewpoint is also related to work on test-time scaling and verifier-guided inference. Chain-of-thought prompting and theoretical analyses of reasoning traces show that inference-time computation can change what a model effectively computes (Kojima et al., 2022; Feng et al., 2023; Phan et al., 2023). Recent studies of adaptive test-time strategies, best-of-$N$, and verifier-guided selection show that verifier quality can alter scaling laws (Wu et al., 2024; Beirami et al., 2025; Setlur et al., 2025; Botta et al., 2025). Worst-case versus average-case guarantees for LLMs have also been studied through verifier-based notions such as verifiability (Amit et al., 2025). That work analyzes learning procedures for self-proving or verifier-backed models. Our focus is narrower and more theorem-proving-specific: given a finite-horizon proof-search process, we expose how action-value error, occupied-region geometry, margins, and average proof length enter the success probability. More abstractly, interactive learning theory has long treated queries and feedback as statistical resources, as in Angluin's $L^*$ algorithm (Angluin, 1987), and recent work on computationally bounded information and formal-language expressivity provides additional vocabulary for reasoning systems (Finzi et al., 2026; Strobl et al., 2024). What is missing from that literature is a theorem-proving-specific bridge from verifier-backed score estimation to end-to-end proof success on structured instance distributions. That is the gap addressed here. Appendix E gives a longer overview.

This interactive-learning connection is more than an analogy. In formal theorem proving, the verifier is an information channel: each tactic attempt returns structured feedback about which branch remains open, which constraints are unsatisfied, and whether an intermediate step was valid. Learning to prove is therefore not only a problem of fitting a static predictor from past data. It is also a problem of learning how to extract useful information from verifier interaction under a limited budget of proof attempts.

**Positioning.** Our contribution is therefore not a survey of theorem proving systems per se, nor a generic theory of test-time scaling. It is a bridge between them: a value-based framework in which proof-theoretic structure, verifier interaction, representation geometry, and learning guarantees on the occupied trace region all appear in the same performance bound.

## 7. Conclusion

We proposed a statistical provability theory for agentic theorem proving centered on a simple pipeline: offline action-value regression followed by greedy proving. The theory keeps the logical notion of provability intact through faithful abstraction, but recasts the evaluation problem in terms of

finite-budget success probability on the problem stream $q_0$. The resulting bound is explicit about what helps: shorter effective proofs, smoother value functions, better covered representations, lower-variance verifier labels, and larger action margins. A useful feature of the result is that one complexity multiplier is directly interpretable from traces: the learned prover's average truncated proof length. This does not remove worst-case hardness. It explains why agentic theorem provers can still work well on the structured theorem distributions that arise in practice.

**Limitations.** The model is intentionally simplified. We analyze greedy proving after offline value regression, while practical systems use beam search, backtracking, retrieval refreshes, proof repair, and multiple interacting agents. The rollout-label assumption is also idealized, and the compact concentration, coverage, Lipschitz, and margin conditions are modeling assumptions about the occupied trace region rather than guarantees about the full proof-search space.

**Future work.** A natural next step is to estimate the quantities in the bound from real prover traces: occupied-region coverage, effective dimension, label variance, action-gap tails, and the unsolved-mass curve. Another direction is to extend the same Bellman-certificate analysis to richer planners such as beam search, top-$k$ search with backtracking, and decomposition policies that explicitly synthesize intermediate lemmas.

## Acknowledgements

SS was supported by JST BOOST JPMJBY24E2, JST CREST JPMJCR25I5, and JSPS KAKENHI 24K21316. YU was supported by JST CREST JPMJCR21M3.

## Impact Statement

This paper presents theoretical work whose goal is to advance the field of Machine Learning by developing a statistical framework for analyzing agentic theorem-proving pipelines. The results are not directly deployable as a system and do not, by themselves, cause immediate harm. However, because our analysis characterizes which pipeline components and design choices most strongly affect success probability, it could be repurposed to improve the effectiveness of agentic models in other domains. In particular, a malicious actor could apply the same principles to engineer more capable harmful agents (e.g., by optimizing search, decomposition, or tool use). We therefore emphasize that our intent is to support transparency and safety-oriented understanding of agentic systems.

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

# A. Additional Technical Details

This appendix gives full proofs for Propositions 2.1 and 2.2, Theorem 3.1, and Corollary 3.1, records sufficient conditions for the optimizer clauses in Assumption 3.1, and supports Sections 2 and 3.

## A.1. Proof of the deterministic-MDP proposition

*Proof of Proposition 2.1.* Define the MDP state space to be $\mathcal{X}$, the action space to be $\mathcal{A}$, the transition kernel to be

$$P(\cdot \mid x, a) = \delta_{F(x,a)},$$

and the goal set to be G. Because the next state depends only on the current state and chosen action, the Markov property holds. Because the kernel is a Dirac mass, the transition is deterministic. Reaching G within $B$ steps is exactly the same event as completing the proof within $B$ verifier calls, so this yields a deterministic finite-horizon reachability MDP. □

## A.2. Proof of the faithful-abstraction proposition

*Proof of Proposition 2.2.* If $\mathrm{Prov}_B(r) = 1$, then by definition there exists a valid proof-search trace or certificate that reaches a closed state within at most $B$ verifier calls. Following the corresponding action sequence in the represented system reaches G within the same budget because the transition and solved-set abstractions are faithful. Hence $V_B^*(e(r)) = 1$.

Conversely, if $V_B^*(e(r)) = 1$, then some action sequence reaches G in the represented system within $B$ steps. Faithful abstraction lifts that sequence back to the lower-level verifier dynamics and solved set, yielding a valid proof-search trace and hence a proof certificate within budget $B$. Therefore $\mathrm{Prov}_B(r) = 1$. □

## A.3. Well-posedness of Bellman and greedy maximizers

**Proposition A.1** (Sufficient conditions for maximizers). *Suppose $\mathcal{X}$ and $\mathcal{A}$ are compact metric spaces, $G \subseteq \mathcal{X}$ is closed, and the deterministic verifier transition $F : \mathcal{X} \times \mathcal{A} \to \mathcal{X}$ is continuous. Then for every finite horizon $t$, the Bellman value $V_t^*$ is upper semicontinuous and, for every state $x$, the supremum*

$$\sup_{a \in \mathcal{A}} Q_t^*(x, a)$$

*is attained by at least one action. Moreover, if $\mathcal{C}_t \subseteq \mathcal{X} \times \mathcal{A}$ is compact, the slice $\mathcal{A}_t(x) := \{a : (x, a) \in \mathcal{C}_t\}$ is nonempty, and $\widehat{Q}_t$ is continuous on $\mathcal{C}_t$, then the greedy maximizer*

$$\hat{\pi}_t(x) \in \arg\max_{a \in \mathcal{A}_t(x)} \widehat{Q}_t(x, a)$$

*exists. Consequently, the optimizer clauses in Assumption 3.1 hold whenever the compact concentration set additionally retains at least one global Bellman maximizer at each relevant state.*

*Proof.* For the Bellman maximizer, use backward induction. The base function $V_0^*(x) = \mathbf{1}[x \in G]$ is upper semicontinuous because G is closed. If $V_{t-1}^*$ is upper semicontinuous, then

$$Q_t^*(x, a) = V_{t-1}^*(F(x, a))$$

is upper semicontinuous on $\mathcal{X} \times \mathcal{A}$, since $F$ is continuous. An upper semicontinuous function attains its maximum on a compact set, so $\sup_{a \in \mathcal{A}} Q_t^*(x, a)$ is attained. The map $x \mapsto \sup_{a \in \mathcal{A}} Q_t^*(x, a)$ is also upper semicontinuous by the maximum theorem, and hence

$$V_t^*(x) = \mathbf{1}[x \in G] \vee \sup_{a \in \mathcal{A}} Q_t^*(x, a)$$

is upper semicontinuous.

For the learned greedy action, the slice $\{x\} \times \mathcal{A}$ is closed in $\mathcal{X} \times \mathcal{A}$, so $\mathcal{C}_t \cap (\{x\} \times \mathcal{A})$ is compact. Its projection $\mathcal{A}_t(x)$ is compact and nonempty by assumption. Since $\widehat{Q}_t$ is continuous on $\mathcal{C}_t$, Weierstrass' theorem gives a maximizer on this slice. The final claim is exactly the candidate-retention condition: one of the global Bellman maximizers must belong to $\mathcal{A}_t(x)$. □

## A.4. Bellman structure and occupancy-sensitive regret

**Proposition A.2** (Deterministic Bellman recursion). *For every remaining horizon $t \geq 1$,*

$$V_0^*(x) = \mathbf{1}[x \in \mathsf{G}], \qquad V_t^*(x) = \mathbf{1}[x \in \mathsf{G}] \vee \sup_{a \in \mathcal{A}} Q_t^*(x, a).$$

*Proof.* If $x \in \mathsf{G}$, then success has already been achieved and $V_t^*(x) = 1$ for every $t \geq 0$. This gives the formula for solved states and, in particular, for $V_0^*$. Now fix $t \geq 1$ and $x \notin \mathsf{G}$. For any policy, the first action $a$ leads to the deterministic next state $F(x, a)$, from which the highest remaining success probability is $V_{t-1}^*(F(x, a)) = Q_t^*(x, a)$. Hence

$$V_t^*(x) \leq \sup_{a \in \mathcal{A}} Q_t^*(x, a).$$

Conversely, for any $\xi > 0$ one can choose an action $a_\xi$ such that

$$Q_t^*(x, a_\xi) \geq \sup_{a \in \mathcal{A}} Q_t^*(x, a) - \xi.$$

Taking $a_\xi$ first and then an optimal continuation policy for the remaining $t - 1$ steps yields

$$V_t^*(x) \geq Q_t^*(x, a_\xi) \geq \sup_{a \in \mathcal{A}} Q_t^*(x, a) - \xi.$$

Sending $\xi \downarrow 0$ proves the claim. □

**Lemma A.1** (One-step greedy regret). *Fix $t \in \{1, \ldots, B\}$ and a relevant state $x$. If*

$$\sup_{a:(x,a) \in \mathcal{C}_t} |\widehat{Q}_t(x, a) - Q_t^*(x, a)| \leq \varepsilon_t,$$

*then*

$$\sup_{a:(x,a) \in \mathcal{C}_t} Q_t^*(x, a) - Q_t^*(x, \hat{\pi}_t(x)) \leq 2\varepsilon_t.$$

*Proof.* Fix any $a$ with $(x, a) \in \mathcal{C}_t$. By greedy selection,

$$\widehat{Q}_t(x, a) \leq \widehat{Q}_t(x, \hat{\pi}_t(x)).$$

Therefore

$$\begin{aligned}
Q_t^*(x, a) &\leq \widehat{Q}_t(x, a) + \varepsilon_t \\
&\leq \widehat{Q}_t(x, \hat{\pi}_t(x)) + \varepsilon_t \\
&\leq Q_t^*(x, \hat{\pi}_t(x)) + 2\varepsilon_t.
\end{aligned}$$

Taking the supremum over $a$ yields the result. □

**Theorem A.1** (Occupancy-sensitive provability bound). *Assume that for every relevant state $x$ and every remaining horizon $t = 1, \ldots, B$,*

$$\sup_{a:(x,a) \in \mathcal{C}_t} |\widehat{Q}_t(x, a) - Q_t^*(x, a)| \leq \varepsilon_t.$$

*Assume also the candidate-retention condition that an action $a_t^*(x)$ exists with $(x, a_t^*(x)) \in \mathcal{C}_t$ and*

$$Q_t^*(x, a_t^*(x)) = \sup_{a \in \mathcal{A}} Q_t^*(x, a)$$

*on those relevant states. Then*

$$\mathrm{SP}_B^{\hat{\pi}}(q_0) \geq \mathrm{SP}_B^*(q_0) - 2\sum_{s=1}^{B} \mathbb{P}_{q_0}^{\hat{\pi}}(T_\mathsf{G} \geq s)\,\varepsilon_{B-s+1}.$$

*Proof.* For $t \in \{0, \ldots, B\}$, define
$$D_t(x) := V_t^*(x) - V_t^{\hat{\pi}}(x).$$

If $x \in \mathsf{G}$, then $D_t(x) = 0$. Now fix $t \geq 1$ and $x \notin \mathsf{G}$. By Proposition A.2,
$$V_t^*(x) = \sup_{a:(x,a)\in\mathcal{C}_t} Q_t^*(x, a),$$

where we used the candidate-retention condition in the theorem statement. Because the learned prover chooses $\hat{\pi}_t(x)$ first and then continues with the same policy,
$$V_t^{\hat{\pi}}(x) = V_{t-1}^{\hat{\pi}}(F(x, \hat{\pi}_t(x))).$$

Since $Q_t^*(x, \hat{\pi}_t(x)) = V_{t-1}^*(F(x, \hat{\pi}_t(x)))$, we obtain
$$D_t(x) = \sup_{a\in\mathcal{A}} Q_t^*(x, a) - Q_t^*(x, \hat{\pi}_t(x)) + D_{t-1}(F(x, \hat{\pi}_t(x))).$$

Applying Lemma A.1 gives
$$D_t(x) \leq 2\varepsilon_t + D_{t-1}(F(x, \hat{\pi}_t(x))).$$

Since $D_t(x) = 0$ on solved states, we may write uniformly
$$D_t(x) \leq 2\varepsilon_t \, \mathbf{1}[x \notin \mathsf{G}] + D_{t-1}(F(x, \hat{\pi}_t(x))).$$

Now run the learned prover from $X_0 \sim q_0$, and let $X_s$ be the state after $s$ executed steps. Applying the inequality above to $X_{s-1}$ with remaining horizon $B - s + 1$ gives
$$D_{B-s+1}(X_{s-1}) \leq 2\varepsilon_{B-s+1}\mathbf{1}[X_{s-1} \notin \mathsf{G}] + D_{B-s}(X_s).$$

Taking expectations and using
$$\mathbb{P}(X_{s-1} \notin \mathsf{G}) = \mathbb{P}_{q_0}^{\hat{\pi}}(T_\mathsf{G} \geq s),$$

we obtain
$$\mathbb{E}[D_{B-s+1}(X_{s-1})] \leq 2\varepsilon_{B-s+1}\mathbb{P}_{q_0}^{\hat{\pi}}(T_\mathsf{G} \geq s) + \mathbb{E}[D_{B-s}(X_s)].$$

Summing this recursion over $s = 1, \ldots, B$ yields
$$\mathbb{E}[D_B(X_0)] \leq 2\sum_{s=1}^{B} \mathbb{P}_{q_0}^{\hat{\pi}}(T_\mathsf{G} \geq s)\,\varepsilon_{B-s+1}.$$

Since $\mathbb{E}[D_B(X_0)] = \mathrm{SP}_B^*(q_0) - \mathrm{SP}_B^{\hat{\pi}}(q_0)$, the result follows. $\qquad\square$

### A.5. Uniform regression error on compact concentration sets

**Lemma A.2** (Sample coverage from the exploration distribution). *Fix $t \in \{1, \ldots, B\}$. Assume that $\mathcal{C}_t$ is compact and*
$$q_t(B(z, r)) \geq c_t r^{d_t}$$

*for all $z \in \mathcal{C}_t$ and sufficiently small $r > 0$. Then there exists a universal constant $C_{\mathrm{cov}} > 0$ such that, with probability at least $1 - \delta_t$, the sampled query pairs $\{(X_{t,i}, A_{t,i})\}_{i=1}^{N_t}$ form an $\eta_t$-net of $\mathcal{C}_t$, where*
$$\eta_t := C_{\mathrm{cov}}c_t^{-1/d_t} \left(\frac{\log(2N_t/\delta_t)}{N_t}\right)^{1/d_t}.$$

*Proof.* Fix $\eta > 0$ small enough that the lower-mass condition applies at scales $\eta/4$ and $\eta/2$. Let $\{z_1, \ldots, z_M\} \subseteq \mathcal{C}_t$ be a maximal $\eta/2$-separated set. Then the balls $B(z_j, \eta/4)$ are pairwise disjoint, and each has $q_t$-mass at least $c_t(\eta/4)^{d_t}$. Since $q_t$ is a probability measure,
$$1 \geq \sum_{j=1}^{M} q_t(B(z_j, \eta/4)) \geq M\, c_t(\eta/4)^{d_t},$$

so
$$M \leq 4^{d_t} c_t^{-1} \eta^{-d_t}.$$

Maximality implies that the balls $B(z_j, \eta/2)$ cover $\mathcal{C}_t$.

For any fixed $j$, the probability that none of the $N_t$ i.i.d. samples falls into $B(z_j, \eta/2)$ is at most
$$(1 - q_t(B(z_j, \eta/2)))^{N_t} \leq \exp\big(-N_t c_t (\eta/2)^{d_t}\big).$$

By a union bound over $j = 1, \ldots, M$, the probability that some covering ball is missed is at most
$$M \exp\big(-N_t c_t (\eta/2)^{d_t}\big) \leq 4^{d_t} c_t^{-1} \eta^{-d_t} \exp\big(-N_t c_t (\eta/2)^{d_t}\big).$$

Choosing
$$\eta = C_{\text{cov}} c_t^{-1/d_t} \left( \frac{\log(2N_t/\delta_t)}{N_t} \right)^{1/d_t}$$

with $C_{\text{cov}}$ sufficiently large makes this upper bound at most $\delta_t$. On the resulting event, every point of $\mathcal{C}_t$ lies within distance $\eta$ of some sampled query pair. $\square$

**Theorem A.2** (Uniform regression error). *Fix $t \in \{1, \ldots, B\}$. Under the depth-wise conditions in Assumption 3.1, for every $\delta_t \in (0, 1)$, with probability at least $1 - \delta_t$,*

$$\|\widehat{Q}_t - Q_t^*\|_{L_\infty(\mathcal{C}_t)} \leq \varepsilon_{\text{app},t} + C_{\text{cov}}(L_{H,t} + L_{Q,t}) c_t^{-1/d_t} \left( \frac{\log(4N_t/\delta_t)}{N_t} \right)^{1/d_t} + 2 \sqrt{ \frac{\log(4N_t/\delta_t)}{2m_t} }.$$

*Proof.* Define
$$\varepsilon_{\text{app},t} := \inf_{h \in \mathcal{H}_t} \|h - Q_t^*\|_{L_\infty(\mathcal{C}_t)}.$$

By Lemma A.2, with probability at least $1 - \delta_t/2$, the sampled query pairs form an $\eta_t$-net of $\mathcal{C}_t$ with
$$\eta_t = C_{\text{cov}} c_t^{-1/d_t} \left( \frac{\log(4N_t/\delta_t)}{N_t} \right)^{1/d_t}.$$

Also, conditional on the sampled query pairs, Hoeffding's inequality implies that for any $\alpha_t > 0$,
$$\mathbb{P}\big( |\bar{Y}_{t,i} - Q_t^*(X_{t,i}, A_{t,i})| > \alpha_t \,\big|\, X_{t,i}, A_{t,i} \big) \leq 2 e^{-2m_t \alpha_t^2}.$$

Setting
$$\alpha_t := \sqrt{ \frac{\log(4N_t/\delta_t)}{2m_t} }$$

and taking a union bound over $i = 1, \ldots, N_t$, we get an event of probability at least $1 - \delta_t/2$ on which
$$\max_{1 \leq i \leq N_t} |\bar{Y}_{t,i} - Q_t^*(X_{t,i}, A_{t,i})| \leq \alpha_t.$$

Work on the intersection of these two events.

Fix $\xi > 0$. By definition of $\varepsilon_{\text{app},t}$, there exists $h_t^\xi \in \mathcal{H}_t$ satisfying
$$\|h_t^\xi - Q_t^*\|_{L_\infty(\mathcal{C}_t)} \leq \varepsilon_{\text{app},t} + \xi.$$

Since $\widehat{Q}_t$ minimizes the maximum empirical absolute deviation,
$$\max_{1 \leq i \leq N_t} |\widehat{Q}_t(X_{t,i}, A_{t,i}) - \bar{Y}_{t,i}| \leq \max_{1 \leq i \leq N_t} |h_t^\xi(X_{t,i}, A_{t,i}) - \bar{Y}_{t,i}|$$
$$\leq \varepsilon_{\text{app},t} + \xi + \alpha_t.$$

Therefore, for each sample point,

$$|\widehat{Q}_t(X_{t,i}, A_{t,i}) - Q_t^*(X_{t,i}, A_{t,i})| \leq |\widehat{Q}_t(X_{t,i}, A_{t,i}) - \bar{Y}_{t,i}| + |\bar{Y}_{t,i} - Q_t^*(X_{t,i}, A_{t,i})|$$
$$\leq \varepsilon_{\mathrm{app},t} + \xi + 2\alpha_t.$$

Now fix any $(x, a) \in \mathcal{C}_t$. Choose a sampled query pair $(X_{t,i}, A_{t,i})$ within distance $\eta_t$. Since both $\widehat{Q}_t \in \mathcal{H}_t$ and $Q_t^*$ are Lipschitz on $\mathcal{C}_t$,

$$|\widehat{Q}_t(x, a) - Q_t^*(x, a)| \leq |\widehat{Q}_t(x, a) - \widehat{Q}_t(X_{t,i}, A_{t,i})| + |\widehat{Q}_t(X_{t,i}, A_{t,i}) - Q_t^*(X_{t,i}, A_{t,i})| + |Q_t^*(X_{t,i}, A_{t,i}) - Q_t^*(x, a)|$$
$$\leq (L_{H,t} + L_{Q,t})\eta_t + \varepsilon_{\mathrm{app},t} + \xi + 2\alpha_t.$$

Taking the supremum over $(x, a) \in \mathcal{C}_t$ and then sending $\xi \downarrow 0$ proves the claim. $\square$

## A.6. Margin theorem and full proof of the main theorem

**Theorem A.3** (Fast rate under a margin condition). *Assume that for every remaining horizon $t$ and every relevant state $x$,*

$$\sup_{a:(x,a)\in\mathcal{C}_t} |\widehat{Q}_t(x, a) - Q_t^*(x, a)| \leq \varepsilon_t$$

*and that the candidate-retention condition in Theorem A.1 holds. Let $\Delta_t(x)$ be the gap between the best and second-best candidate action values at remaining horizon $t$. If the conditional occupancy distributions $O_s^{\hat{\pi}}$ satisfy*

$$O_s^{\hat{\pi}}(\Delta_t \leq u) \leq C_\Delta u^\beta, \qquad t = B - s + 1,$$

*then*

$$\mathrm{SP}_B^{\hat{\pi}}(q_0) \geq \mathrm{SP}_B^*(q_0) - 2C_\Delta \sum_{s=1}^{B} \mathbb{P}_{q_0}^{\hat{\pi}}(T_\mathsf{G} \geq s)(2\varepsilon_{B-s+1})^{\beta+1}.$$

*Proof.* For elapsed step $s \in \{1, \ldots, B\}$, let $t = B - s + 1$ and define

$$r_s(x) := \max_{a:(x,a)\in\mathcal{C}_t} Q_t^*(x, a) - Q_t^*(x, \hat{\pi}_t(x)).$$

By Lemma A.1, $r_s(x) \leq 2\varepsilon_t$. Moreover, if $\Delta_t(x) > 2\varepsilon_t$, then the perturbation $\widehat{Q}_t - Q_t^*$ is too small to change the identity of the optimal action, so $r_s(x) = 0$. Hence

$$r_s(x) \leq 2\varepsilon_t \, \mathbf{1}[\Delta_t(x) \leq 2\varepsilon_t].$$

Conditioning on $T_\mathsf{G} \geq s$, the state $X_{s-1}$ has distribution $O_s^{\hat{\pi}}$. Therefore

$$\mathbb{E}[r_s(X_{s-1})\mathbf{1}[T_\mathsf{G} \geq s]] = \mathbb{P}_{q_0}^{\hat{\pi}}(T_\mathsf{G} \geq s) \int r_s(x) \, O_s^{\hat{\pi}}(\mathrm{d}x)$$
$$\leq 2\varepsilon_t \, \mathbb{P}_{q_0}^{\hat{\pi}}(T_\mathsf{G} \geq s) O_s^{\hat{\pi}}(\Delta_t \leq 2\varepsilon_t)$$
$$\leq 2C_\Delta \, \mathbb{P}_{q_0}^{\hat{\pi}}(T_\mathsf{G} \geq s)(2\varepsilon_t)^{\beta+1}.$$

Summing over $s = 1, \ldots, B$ and using the same telescoping argument as in Theorem A.1 proves the theorem. $\square$

*Full proof of Theorem 3.1.* For each $t = 1, \ldots, B$, Theorem A.2 gives an event $\mathcal{E}_t$ of probability at least $1 - \delta_t$ on which

$$\sup_{(x,a)\in\mathcal{C}_t} |\widehat{Q}_t(x, a) - Q_t^*(x, a)| \leq \varepsilon_t,$$

with $\varepsilon_t$ exactly as defined in Equation (3). By a union bound, the event

$$\mathcal{E} := \bigcap_{t=1}^{B} \mathcal{E}_t$$

has probability at least $1 - \sum_{t=1}^{B} \delta_t$. On $\mathcal{E}$, Theorem A.1 implies Equation (2). If the margin condition holds, then Theorem A.3 implies Equation (4). This proves the theorem. $\square$

*Proof of Corollary 3.1.* If $\varepsilon_t \leq \bar{\varepsilon}_B$ for all $t$, then Equation (2) immediately gives

$$\mathrm{SP}_B^{\hat{\pi}}(q_0) \geq \mathrm{SP}_B^*(q_0) - 2\bar{\varepsilon}_B \sum_{s=1}^{B} \mathbb{P}_{q_0}^{\hat{\pi}}(T_{\mathsf{G}} \geq s).$$

The tail-sum identity yields

$$\sum_{s=1}^{B} \mathbb{P}_{q_0}^{\hat{\pi}}(T_{\mathsf{G}} \geq s) = \mathrm{Len}_B^{\hat{\pi}}(q_0),$$

which proves the first claim. For the conditional statement, Proposition 2.2 implies

$$\mathrm{SP}_B^*(q_0) = q_0(\mathrm{Prov}_B = 1).$$

Also, if $\mathrm{Prov}_B(x) = 0$, then $V_B^{\hat{\pi}}(x) = 0$. Hence

$$\mathrm{SP}_B^{\hat{\pi}}(q_0) = q_0(\mathrm{Prov}_B = 1)\, \mathbb{E}_{q_0}\left[V_B^{\hat{\pi}}(X) \mid \mathrm{Prov}_B(X) = 1\right].$$

Combining this identity with the first bound yields the result. □

## B. Variable Number of Goals and Truncation

This appendix expands the variable-goal and compactness discussion in Sections 2 and 4.

The measure-valued representation naturally handles a variable number of open goals, but a globally compact state space may require a mass cap. Let $M_{\leq W}(K)$ be the set of finite positive measures on a compact goal-embedding space $K$ with total mass at most $W$. Define the overflow event

$$\mathsf{Overflow} := \left\{ \max_{t \leq B} X_t(K) > W \right\}.$$

The cap $W$ should not be read as a claim that real proof states have a uniformly bounded number of subgoals. It is only a technical device for moving from a high-probability region of the proof-search process to a globally compact surrogate. The excluded trajectories are exactly those on which the active goal multiset becomes too large under the policy and budget being analyzed.

**Proposition B.1** (High-probability truncation). *Let $V_{B,\leq W}^{\pi}$ denote the success probability in the truncated system that sends overflow states to an absorbing failure state. If $\mathbb{P}_x^{\pi}(\mathsf{Overflow}) \leq \delta_W$, then*

$$V_B^{\pi}(x) \geq V_{B,\leq W}^{\pi}(x) - \delta_W.$$

*Proof.* Couple the true and truncated processes until the first overflow time. They coincide on the complement of $\mathsf{Overflow}$. Since the truncated process only removes successful trajectories after overflow, the difference in success probability is at most the overflow probability. □

This is why the main text only assumes compact concentration sets: global compactness can be recovered by truncation plus an additive tail term. In applications, $W$ can be chosen from empirical proof traces or from a conservative planner budget; the theorem then reports the price of ignoring wider proof states through the single tail probability $\delta_W$.

## C. Representation Learning Through Certificate Tightness

This appendix expands the Bellman-certificate discussion in Section 4.

The theorem suggests a concrete objective for representation learning. Let $\Phi$ be a class of embeddings $\phi$ that map formal goals into a metric space $K_\phi$. Each $\phi$ induces a state representation, a concentration set $\mathcal{C}_{\phi,t}$, and hypothesis classes $\mathcal{H}_{\phi,t}$. One can then choose $\phi$ to balance approximation and geometry:

$$\min_{\phi \in \Phi} \mathbb{E}_{q_0}\left[U_{B,\phi}^*(X_0) - L_{B,\phi}^*(X_0)\right] + \lambda\, \mathsf{Comp}(\phi),$$

where $U_{B,\phi}^*$ and $L_{B,\phi}^*$ are the tightest Bellman-based upper and lower certificates available inside a restricted function class, and $\mathsf{Comp}(\phi)$ penalizes statistically unfriendly geometries such as large covering dimension or extreme Lipschitz distortion.

This viewpoint formalizes a common intuition: a good representation is not merely predictive, but one under which provability certificates are tight and statistically estimable. The certificate gap $U_{B,\phi}^*(X_0) - L_{B,\phi}^*(X_0)$ measures how much uncertainty remains about finite-budget provability after restricting attention to the surrogate family induced by $\phi$. The complexity penalty then prevents choosing a representation that makes certificates expressive but statistically impossible to estimate from verifier-backed traces. This criterion is deliberately aligned with Theorem 3.1: the same representation should reduce approximation error, improve local coverage, and preserve the action margins that make greedy proving stable.

## D. Scaling-Law Interpretation

This appendix expands the scaling-law viewpoint in Sections 4 and 5.

Ignoring logarithmic factors and approximation error, the depth-wise statistical error behaves like

$$\varepsilon_t \approx n_t^{-1/(d_t+2)}.$$

Plugging this into Equation (2) gives a schematic sample-complexity relationship

$$\mathrm{SP}_B^*(q_0) - \mathrm{SP}_B^{\hat{\pi}}(q_0) \lesssim \sum_{s=1}^{B} \mathbb{P}_{q_0}^{\hat{\pi}}(T_\mathsf{G} \geq s)\, n_{B-s+1}^{-1/(d_{B-s+1}+2)}.$$

Under a margin condition, the exponent improves to $(\beta+1)/(d_t+2)$.

Three qualitative scaling levers become explicit. First, if decomposition or lemma introduction reduces the average proof length, the probabilities $\mathbb{P}_{q_0}^{\hat{\pi}}(T_\mathsf{G} \geq s)$ decay earlier and the sum has fewer large terms. Second, if retrieval or representation learning localizes the relevant state-action pairs to a lower-dimensional region, then the effective dimension $d_t$ decreases and the rate in $n_t$ improves. Third, if verifier feedback and better scoring make near-ties rarer, the margin exponent $\beta$ increases and the fast-rate regime becomes more favorable. These levers correspond to different engineering interventions, but the bound puts them on the same scale: each reduces the value-estimation error accumulated along the learned prover's occupied trajectory. This is the statistical sense in which agentic components can produce large practical gains without changing worst-case logical hardness.

## E. Extended Related Work

This appendix extends the related-work discussion in Section 6.

**Before the LLM era: tactic and premise guidance.** Long before current reasoning models, theorem proving research already recognized two bottlenecks: choosing useful premises from a large library and choosing the next proof step so that symbolic search does not explode combinatorially. TacticToe (Gauthier et al., 2017; 2021) is representative: it learns tactic guidance from existing proofs and combines those predictions with Monte Carlo tree search. HOList and DeepHOL (Bansal et al., 2019) established a large-scale environment for higher-order theorem proving and made reinforcement-learning and imitation-learning approaches easier to compare. These systems already fit the MDP perspective quite naturally: proof states evolve under discrete actions, learning affects search bias, and evaluation is budgeted success rather than logical completeness.

**Datasets and environments.** Infrastructure changed the field by making proof interaction reproducible. LeanDojo (Yang et al., 2023) exposed Lean proof states, tactics, and premise annotations through a programmatic API, which made retrieval-augmented and verifier-interactive learning much more practical. miniF2F (Zheng et al., 2022) provided a cross-system benchmark built around olympiad-style mathematics and became a common testing ground for neural and LLM-based formal provers. This benchmark orientation is relevant to our theory because it sharpened the need for a distributional notion of success rather than a purely worst-case one.

**Neural language models for formal proof.** GPT-f (Polu & Sutskever, 2020) was an early signal that generative language models could meaningfully contribute to formal proof search. The shift from fixed handcrafted features to learned sequence

or graph representations changed both training and inference. Rather than only predicting the next rule locally, models could propose longer fragments, re-rank retrieved lemmas, and use verifier feedback to refine failed attempts. This made theorem proving look less like conventional search with a static heuristic and more like a dynamic loop of proposal, verification, and revision.

**Agentic provers and the decomposition trend.** Modern systems increasingly treat theorem proving as a structured workflow with multiple interacting components. DSP (Jiang et al., 2023) uses informal proof sketches to steer formal search toward easier subproblems. DeepSeek-Prover and DeepSeek-Prover-V2 (Xin et al., 2024; Ren et al., 2025) emphasize training data generation, recursive decomposition, and RL for Lean 4 proving. Prover Agent (Baba et al., 2025) explicitly separates roles such as informal reasoning and formal proving. Seed Prover (Chen et al., 2025) centers lemma-style whole-proof reasoning with iterative refinement using Lean feedback. Hilbert (Varambally et al., 2025) and Aristotle (Achim et al., 2025) also combine informal reasoning, explicit intermediate lemmas, and proof-assistant checking. The common pattern is that decomposition and lemma reuse are no longer auxiliary heuristics; they are first-class levers for changing the effective proof geometry.

**Test-time scaling and verifiers.** Chain-of-thought prompting (Kojima et al., 2022) and subsequent theory (Feng et al., 2023; Phan et al., 2023) highlighted that inference-time computation is part of the effective hypothesis class. In parallel, recent work has studied test-time scaling more broadly, including how performance depends on adaptive allocation of compute (Wu et al., 2024). Best-of-$N$ and verifier-guided inference analyses (Beirami et al., 2025; Setlur et al., 2025; Botta et al., 2025) emphasize that verifier quality can change scaling laws. In theorem proving, the proof assistant is a nearly perfect verifier for formal correctness, but agents still rely on imperfect learned signals for premise usefulness, subgoal quality, or semantic progress. Our theory is meant to capture this interaction between perfect syntactic checking and imperfect statistical scoring.

**Interactive learning and computational constraints.** From a broader ML perspective, theorem proving sits inside a long line of work where learning is driven by interaction rather than passive i.i.d. examples. Angluin's $L^*$ algorithm (Angluin, 1987) is a canonical example: the learner uses membership and equivalence queries to identify a target automaton. Recent work such as epiplexity (Finzi et al., 2026) and surveys on formal languages and transformer expressivity (Strobl et al., 2024) further emphasize that computational constraints matter when translating information into performance. Our position is that formal theorem proving requires both viewpoints at once: proof-theoretic structure determines what short successful trajectories exist, while statistical learning determines whether an agent can identify and exploit those trajectories from realistic data.

