# OpenReview forum: "Why Agentic Theorem Prover Works: A Statistical Provability Theory of Mathematical Reasoning Models"
_ICML.cc/2026/Conference — ICML 2026 regular_

### Official Review · Reviewer_ijpq · 2026-03-02

**Soundness:** 3
**Presentation:** 2
**Significance:** 3
**Originality:** 3
**Overall Recommendation:** 4
**Confidence:** 3

**Summary:**

This paper tries to formalize the difference between average-case and worst-case hardness when it comes to modern agentic theorem provers, and in particular to explain empirical success in cases where the worst-case instance is hard. They ask what changes when a problem needs only to be solved on a certain structured distribution and not in the worst-case. Their hypothesis is that real-world distributions of theorems concentrate on a highly structured and repetitive subset of mathematics. The authors model theorem proving as a finite-horizon reachability problem in a Markov decision process (MDP), where states represent proof obligations, actions represent the next proof step and a transition is the proof assistant’s execution.  They introduce "statistical provability", defined as the time-bounded probability of reaching a verified proof under a given instance distribution. Using the Bellman structure of the MDP, they prove the existence of optimal policies and prove bounds on the performance of score-guided planning methods (e.g. greedy or beam search). Their analysis shows how components such as verifier feedback, retrieval, subgoal decomposition and representation geometry affect success probability through quantities like approximation error, statistical complexity, margins and proof length.

**Compliance With Llm Reviewing Policy:**

Affirmed.

**Final Justification:**

I found this paper technically interesting and potentially useful as a first theoretical framework for thinking about average-case success in modern agentic theorem proving. My main concern was that the theory felt somewhat decoupled from what can currently be validated on real prover traces, together with presentation issues that made some of the technical flow hard to follow. The rebuttal addressed these concerns well, and was appropriately transparent about the current empirical limitations while committing to improve the exposition. Overall, the rebuttal resolved my main questions, and I support acceptance.

**Key Questions For Authors:**

1) Which quantities in your bounds (as details in the weaknesses section) do you believe are measurable on real prover traces, and do you have empirical estimates for any of them?

2) Can you please elaborate on the proof of Theorem 9.1? In particular, are there additional assumptions needed to justify the stated uniform sup-norm bound? And how would you obtain targets Y with Ex[Y | mu, a] = Q_b^*(mu, a)?

3) how is the "relevant domain" defined and what conditions ensure it is stable across runs/model changes?

**Limitations:**

yes

**Strengths And Weaknesses:**

*Strengths*

(-) The paper cleanly formalizes agentic theorem proving as a finite-horizon reachability MDP and introduces statistical provability as the distribution-averaged success probability under a compute budget, aligning well with how these systems are evaluated in practice. It uses mathematical tools for supporting this modeling.

(-) Leveraging the Bellman recursion, the paper proves existence of optimal finite-horizon policies under mild conditions and derives provability upper and lower bounds , and by that, offers a concrete way to reason about provability without fully solving the MDP.

(-) The paper gives a concrete taxonomy of easy instances (short proofs, low geometric complexity, low approximation error) versus harder instances (long proofs, ambiguous states, OOD). This is a very helpful conceptual bridge from abstract bounds to practitioners' intuition.


*Weaknesses*

(-) The strongest weakness in my opinion is that the theory is somewhat decoupled from what can be validated in real theorem-proving. The bounds depend on quantities like the domain geometry (doubling/covering numbers), uniform score approximation over a "relevant" domain or margin tail under an occupancy distribution, however, the paper does not show how to estimate these in practice (or shows empirically that the assumed regimes hold for real provers).

The presentation can be improved, below are a few points on this manner.

(-) I think the abstract assumes some context that not all readers have, I suggest revising it to be more approachable.

(-) I think that adding a few citations to support your arguments in the introduction can be useful. (I also prefer including the related work in the main body rather than in the appendix). One work that may be related is "A Theory for Worst-Case vs. Average-Case Guarantees for LLMs"
I think that they also define a closely related term of "statistical provability" and call it "Verifiability", with respect to a verifier (it also seems to relate to section A.5 and to other parts in your work, and in general the discussion of worst-case vs. average-case). So I recommend checking it and add if relevant.

(-) A few minor editorial comments: when the probability is over, say, \pi, I recommend writing it as \Pr_\pi instead of above the "Pr" notation. I also think it's more standard to emphasize with italic instead of bold. In page 8, the equation after the line "A representative schematic lower bound is:" overflows to the second column. Moreover, I recommend uniting some sections into one section with subsections (because 12 sections for the body of the paper is non-standard). For example, section 5 is very short, and does need stand on its own, I think its better to combine it with other results.

(-) I think it can be helpful to explain what Bellman structure is. This is one example of a wider issue, that you dive into the technical details too fast, which makes it hard to follow, and even in the technical parts some of the proofs are proof sketches which are sometime hard to follow.

(-) The references Lemma A.1 and Lemma A.2 do not appear (appendix D).

---

> ### Author Rebuttal · Authors · 2026-03-28
>
> Thank you for the positive and constructive feedback. We are glad that you found the problem important and that the paper’s attempt to connect theorem-proving practice with a principled value-function view seemed potentially useful. We will reflect your detailed comments in the revised manuscript.
>
> **Q1. Measurable quantities.**  We agree that the current submission does not make the empirical side sufficiently explicit. The final provability bound/evaluation in the paper (the RHS of the inequality on p.8, l.389) is in principle approximable from training data. More concretely: the “relevant domain” $D$ can be approximated by the support of the available training traces; quantities such as $L_{eff}$ and the margin-tail parameter $\alpha$ can be estimated from teacher-LLM trajectories $\{ x_0,a_0,\ldots,x_L\}$; the doubling dimension $d_D$ can be upper bounded by the dimension of the embedding space of LLM; and the covering number of the score class can be related to standard capacity measures from deep learning theory. We do not yet have empirical estimates for these quantities, and we agree that this is an important next step. In the revised manuscript, we will state much more clearly which terms are directly estimable from traces, which are only upper bounded by structural surrogates, and which remain latent modeling assumptions.
>
> **Q2. Theorem 9.1.** We apologize that the appendix/proof pointer was omitted in the submitted version. In the revised manuscript, we will explicitly include the proof, which has three main ingredients: (1) a uniform deviation bound for adaptively collected, non-i.i.d. data, obtained by combining bounded martingale-difference concentration with an $\varepsilon$-net argument; (2) a metric-entropy bound for the Lipschitz score class on the relevant domain $D$, using the doubling properties of $D$; and (3) an oracle/ERM step combining the statistical deviation term with the approximation error. This yields the stated sup-norm control up to logarithmic factors. To justify the bound, one needs the regularity assumptions already implicit in the theorem: bounded targets, the conditional-unbiasedness relation $E[Y\mid \mu,a]=Q_b^\ast(\mu,a)$, and entropy control of the hypothesis class on $D$. As for the targets $Y$, the idealized theorem uses rollout-based bounded success labels whose conditional mean equals $Q_b^\ast$; in practice, these can be approximated by repeated rollouts or related value-target constructions.
>
> **Q3. relevant domain $D$.** Our intention is that $D$ corresponds to the support (more precisely, the non-negligible occupancy region) of $Q_b^\ast(x,a)$. At the theory level this can simply be assumed; at the data level one can replace it with the empirical support $\hat D$ induced by the training traces. Stability across runs/model changes is therefore not automatic: it requires that the new runs remain concentrated on essentially the same occupied region. Under standard concentration assumptions---for example, independent samples and a subgaussian occupancy distribution---one expects $\hat D$ to approximate the underlying occupied region at about the usual $1/\sqrt n$-scale, modulo the chosen metric/proxy.
>
> **Weaknesses**
>
> Thank you also for the detailed presentation comments. We agree that the exposition currently enters technical details too quickly. In particular, by “Bellman structure” we mean the dynamic-programming recursion for the optimal $B$-step success probability and the associated one-step action values $Q_b^\ast$; this is the structural fact that makes the upper/lower certificates and the search-error bounds possible. We will explain this earlier and more explicitly, expand several proof sketches, and improve the proof pointers. The references to Lemma A.1 and Lemma A.2 in Appendix D are indeed typos and should read D.1 and D.2; thank you for catching this.
>
> Finally, thank you for pointing us to the recent related work [1]. We will add it to the revised manuscript and clarify the distinction in contribution. That work studies how to learn “self-proving” models under specific training procedures, including transcript-based learning and verifier-feedback RL, and provides learning-theoretic guarantees for those procedures. Our paper instead studies time-bounded statistical provability under a simpler score-guided theorem-proving abstraction. The contribution we want to emphasize is that the resulting bounds expose data-dependent complexity terms (such as effective proof length / horizon, approximation error, margins, and domain geometry) that can help diagnose where an agent is failing and how improvements in search/representation may change provability.
>
> [1] Amit et al., A Theory for Worst-Case vs. Average-Case Guarantees for LLMs, NeurIPS2025

---

> > ### Author Rebuttal · Reviewer_ijpq · 2026-04-02
> >
> > Thank you for the rebuttal. My concerns are fully resolved:  I appreciate the clarifications regarding the quantities in the bounds that are intended to be estimable from prover traces, the additional explanation of the proof strategy behind Theorem 9.1.
> > I also appreciate the authors' transparency about the current limitations of the empirical side, as well as their commitment to improving the presentation. Overall, the rebuttal addressed my main questions.

---

### Official Review · Reviewer_pzmG · 2026-03-08

**Soundness:** 2
**Presentation:** 1
**Significance:** 2
**Originality:** 1
**Overall Recommendation:** 2
**Confidence:** 5

**Summary:**

Contributions (reformulated):

	1. Formalization of agentic theorem provers as a time-bounded reachability problem in a Markov decision process.
	2. Establish provability certificates.
	3. Demonstrate how proof complexity is capable of producing exponential changes in scaling.
	4. Using cuts to shorten the proofs.

**Compliance With Llm Reviewing Policy:**

Affirmed.

**Final Justification:**

The intense discussion culminated in a debate over terminology and mathematical rigor, which, unfortunately, only reinforced the prior assessment. The authors addressed some issues and questions, but the main ones remain. The presentation of material suffered from familiar shortness of explanations, where a more elaborate argument is necessary, and vice versa. That revealed a lack of necessary rigor in the proofs.

As a result, the evaluation remains the same.

**Key Questions For Authors:**

Q0: Definition 5.1. Why does the supremum always exist? Or is this an assumption? Later, we learn it is, under most tight conditions. Could you clarify?

	Q1. What is the exact provability notion discussed in the paper? If it is Lean cuts, that does not cut it (pun not intended).

	Q2. What does it mean, exactly: "agentic theorem prover works?" The more we read through "Statistical Provability Theory of Mathematical Reasoning Models", the more it becomes an informal and potentially ill-formed expression.

	Q3. Since the considerations are statistical, what probability space are we talking about, in general? That of MDP, General Borel state space, with transition kernel?

	Q4. Why is so much space devoted to Lean? None of this information is used in the main text.
Missed opportunity to simplify into a        one-liner to compactify the cut proof (cut_style):

 theorem cut_style_simp (A B : Prop) (hA : A) (f : A → B) : B ∧ B :=
⟨f hA, f hA⟩

        But the only question is: "Where is this needed in proving any of the theorems?"  If it is Lean cut, that does not cut it (pun not intended).

	Q5. In general, the Appendix contains a lot of unrelated material. Why?

	Q6. What is assumption 6.1? It does not exist as such. Presumably, it is one of many conditions that imply the applicability of the Portmanteau theorem on the weak convergence of a sequence of probability measures. Section "6" on page 4 is also a dangling pointer.

	Q7.  In which result do we establish that agentic provers work exactly? Because, informally, according to Section 3.1, we jump from a general agentic context to a proof assistant (that is a stretch since that is already an HFRL assumption), and only estimate the probability of the next step reaching a solved state within a fixed time/step budget. That is an entirely different problem.

	Q8. Claim on page 8: "Overall, the theory explains why modern agentic systems can be effective (Reviewer: the term is not defined)  on biased, structured theorem distributions (?) without contradicting classical hardness, and it highlights where limitations remain inevitable under uniform or adversarial instance regimes."  Where was that shown?

**Limitations:**

The authors provide a careful section on the impact statement.

**Strengths And Weaknesses:**

$Strengths$ The authors raise an important theme in a multi-view of a few modern technologies: agentic provers, provability in general, and plausible effectiveness of statistical methods in generating proofs. They claim to develop a statistical (a more appropriate term would be stochastic) theory that explains multiple phenomena across this cross-view.

${Weaknesses}$ That increases the approach's complexity. Since it is a rigorous context, all definitions and interpretations must be explicitly justified; the theorem must be strongly logically associated with qualitative claims. That has not been achieved fully. Moreover, the authors brought a lot of unnecessary machinery into the context. As a result, the presentation suffers.

If one parses the title of the paper literally (which is the only reasonable interpretation), then the conclusion is that what we have here is a statistical theory proving that an agentic theorem prover $"\it{works}"$.
Statistically, any automated prover (or verifier) of certain quality works in the sense that it helps generate a valid statement or check the validity of a theorem.

Why then does it have to be agentic necessarily? Are there some advantages/necessities associated with? Or is it simply conducive to the overall placement of AI provers in business infrastructure, etc.?

The reader would be surprised. The authors do not raise any of those questions. They are more concerned to give us an abstraction of an agentic prover such that, under certain (not necessarily justified or natural) conditions, behaves in a prescribed manner. The question of why the agentic theorem prover works as above is not really answered.

Provability is the rigorous term in logic, and the authors do not follow it. As a result, an immediate expectation of having a rigorous result like "agentic architecture provides a reasonable expectation of successful mathematical reasoning," or a probabilistic version of that, is not fulfilled. The authors do provide well-known snippets on Lean, Logic, and Analysis; the main contribution claimed to be "A statistical Provability theory of mathematical reasoning models," but most of the materials in the appendices, while sometimes interesting, are not related to the core of the paper and superficial to the context.

What is a mathematical reasoning model? It appears to be just an AI LRM.
In the abstract, the authors essentially say that LRMs "have recently achieved striking empirical success, yet it remains unclear which components drive performance and why such systems work at all despite classical hardness of proof search". Again, the authors do not explain that point immediately. Instead, they propose a statistical viewpoint in which they introduce a notion of statistical provability defined as the finite-horizon success probability of reaching a verified proof.

The introduction contains sentiments like these: "Real-world theorem instances are not distributed uniformly over 'all statements of a given size.'" Instead, they concentrate on a highly structured, repetitive subset of mathematics (libraries, idioms, canonical
proof patterns." What probability space are we talking about representing those theorem instances, exactly?

So the reader eventually gets an answer. But the exposition continues. The authors formulate a statistical provability hypothesis. Basically, the real-world instances of the theorem are not distributed uniformly, as we saw. At this point, one wonders what probability space we are talking about, again?

Let's now try to answer these questions by reading the paper's body. Here is what we learn.
An agentic theorem prover is a Markov decision process. The provability of an instance $x_0$ under budget (#verifier calls) is

${V^*}_B(x_0) = sup_{\pi} {Pr}^{\pi} (\exists t <= B : X_t \in G | X_0= x_0)$.

This immediately raises many questions, which can be found in the Section "Key questions for authors". This should explain the following

$Conclusion$:
In this current form, one cannot possibly accept the paper. Qualitative claims in this paper do not correspond to the claims made with any attempt at rigor. The authors did not make enough effort to present the material cohesively.
As a result, the paper's soundness suffered as well. A loose attitude towards terms that have a rigorous, often critical meaning is very frequent in this paper. One cannot claim qualitative results of higher strength than what has been rigorously proven. No appendices compensate for the gap between the proven and unproven. In the Section "Key Questions", the authors can find many concerns about their presentation of their results and the apparent lack of effort to explain their points in a discernible way.

---

> ### Author Rebuttal · Authors · 2026-03-28
>
> Thank you for the detailed reading and for pressing us on definitions, scope, and presentation. We agree that the current manuscript does not separate the formal claims from their interpretation clearly enough. In the revised manuscript, we will clarify the core notions, fix the dangling references, and substantially shorten the appendix so that only material needed for the main argument remains.
>
> **Q0. Definition 5.1.** Definition 5.1 introduces the object of study ($V_B^\pi$) and its supremum over policies ($V_B^\ast$). The substantive technical issue is not the formal existence of the least upper bound, but whether this value is **attained** by an optimal policy. Section 6 shows, under mild compactness/continuity assumptions, that the Bellman maximization is attained and that optimal deterministic Markov policies exist. We will state this more explicitly in the revised manuscript.
>
> **Q1. What notion of provability is used?** In mathematical logic, once a theory $\Gamma$ and inference rules are fixed, a proposition $A$ is provable (denoted $\Gamma \vdash A$) if there exists a derivation/proof tree for $A$. Equivalently, one may view this provability as reachability in a hypergraph whose nodes are proof states (e.g., sets of goals/sequents) and whose hyperedges are inference steps. Our paper does not redefine logical provability. Rather, it introduces a **statistical** layer on top of logical provability: after endowing the proof-search process with a transition kernel, we study the probability that a policy reaches a solved state within a fixed budget. This is what we call **statistical provability**.
>
> **Q2. What does “agentic theorem prover works” mean?** In our formalism, it means that the prover finds a verified proof, i.e., reaches the solved set, with nontrivial/high probability within a fixed time/step budget. This is precisely the time-bounded success probability in Definition 5.1. The motivating puzzle is that naive stochastic search in a huge proof space should have extremely small success probability, yet modern AI-based theorem provers often succeed on structured benchmarks. Our paper studies which quantities can make such success possible.
>
> **Q3. What probability space is used?** As stated in Section 3.3, the transition kernel is a Borel probability measure on the proof-state space $\mathcal X$. Since $\mathcal X$ is assumed to be a compact metric space, it carries its Borel $\sigma$-algebra, and the controlled process is a Markov decision process on a general Borel state space. We will make this more explicit in the revised manuscript.
>
> **Q4/Q5. Why so much Lean / appendix material?** Appendix B used Lean and sequent-calculus examples only to illustrate why theorem proving can be viewed as an MDP. Appendix C discussed cut elimination, but this is not used in the main results. We agree that the appendix currently contains material that is not sufficiently tied to the core argument. In the revised manuscript, we will remove appendix material that is not used in the main text and keep only the minimum needed to motivate the abstraction.
>
> **Q6. Assumption 6.1**  Assumption 6.1 is the continuity condition on the transition kernel stated on p.4: intuitively, small perturbations of the proof state $\mu$ and tactic $a$ do not change the next-state distribution abruptly.
>
> **Q7/Q8. Where do we show that agentic provers “work”?** We do **not** prove that agentic provers are universally successful. The formal claim is narrower: we model an agentic theorem prover as a general policy $\pi$ over proof states/actions and derive lower bounds / certificates for its time-bounded proof success probability. These lower bounds become nontrivial when quantities such as effective proof length, score approximation error, coverage, and the action-gap parameter $\alpha$ are controlled. These quantities reflect the complexity of the theorem distribution. Over unrestricted, uniform, or adversarial families, they can diverge or margins can collapse, making the lower bound vacuous; over biased, structured distributions, they remain finite, yielding a principled explanation of why stochastic proof search can succeed without contradicting classical hardness. In the revised version we will state this interpretation more explicitly and tie it more directly to the earlier theorems.

---

> > ### Author Rebuttal · Reviewer_pzmG · 2026-04-02
> >
> > $(Following the \ numeration \  of \ the \ last \ rebuttal)$.
> >
> > Q0. It seems semantics, but your own explanation invalidates the definition. If $sup$ does not exist, the definition is invalid. Attained means "succeeds", in our context, $\textit{sometimes succeeds}$. The definition should then explicitly state the fact, or it would confuse the audience. It is also interesting when authors sometimes say this is essentially equivalent to the original definition. Well, the correct definition is not equivalent to a wrong one. Also, one may consider this a fine (pedantic) point.
> >
> > Q1. Partially addressed. To make it work, we need to specify which logic we work in. Is it a first-order logic of finite graphs? By the same token, which probability space? Is it a standard Borel space with standard measure? Otherwise, saying "we define statistical provability on top of logical one" carries no rigorous meaning. It certainly appeals to a Computer Scientist and a mathematician, too, but the approach would not be sound until the questions above are properly settled. One of the questions is partially answered in Q3.
> >
> > Q2. If the prover finds a verified proof, we are done. What would be a practical use of the notion?
> >
> > Q3. Partially addressed. Why is the state $\mathcal X$ "assumed" compact? That needs to be explained in detail.
> >
> > Q4/5 Addressed.
> >
> > Q6. That may be an empty class for some distributions, kernels, $\mathcal \mu$,  and $\mathcal \alpha$.
> >
> > Q7/Q8. This explanation needs a lot of work. It is still quite vague.

---

> > > ### Author Response · Authors · 2026-04-03
> > >
> > > **Q0.**
> > >
> > > We respectfully ask the reviewer to read the definitions carefully.
> > >
> > > > If sup does not exist, the definition is invalid.
> > >
> > > Please look carefully at Definition 5.1. It introduces two notions of provability: $V_B^\pi$ and $V_B^\ast$. Only one of them involves a supremum; the other does not.
> > >
> > > Next, please look carefully at Theorem 6.1. It shows that the supremum, namely $V_B^\ast$, is attained under very mild conditions.
> > >
> > > Moreover, please see the discussion in Section 8, especially Theorem 8.2. It shows that $V_B^\pi$ remains within a finite gap of $V_B^\ast$.
> > >
> > > > Attained means "succeeds".
> > >
> > > We do not make that statement.
> > >
> > > > authors sometimes say this is essentially equivalent to the original definition
> > >
> > > We do not make that statement either. Statistical provability and provability in logic are different notions.
> > >
> > > **Q1.**
> > >
> > > > Is it a first-order logic of finite graphs?
> > >
> > > Yes, first-order logic (FOL) is included. This is because FOL can be described in sequent calculus, and sequent calculus is a 1-Markov Decision Process (1-MDP), as explained in Section 3.3.
> > >
> > > Our framework includes not only FOL, but also propositional logic and interactive theorem proving systems such as Lean. All of these can be viewed as 1-MDPs.
> > >
> > > > By the same token, which probability space? Is it a standard Borel space with standard measure?
> > >
> > > The underlying measurable space is the standard Borel space $(\mathcal X, \mathcal B(\mathcal X))$, where $\mathcal X = M_{\le W}(K)$ is the metric space introduced in Section 3.3.
> > >
> > > As for the data, we do not require any more specific structure than that it is supported on a compact domain $\mathcal D \subset \mathcal X \times \mathcal A$. Since our analysis is based on uniform approximation of the score function on $\mathcal D$, no additional specification is needed.
> > >
> > > **Q2.**
> > >
> > > In this work, we theoretically evaluate the **proof success probability**. As discussed in Section 11, interpreting the complexity terms that appear in this evaluation can help improve AI systems in practice.
> > >
> > > For example, our bounds indicate that if the proofs appearing in the training data are longer on average, then the proof success probability decreases. This suggests a practical implication: when training AI systems for theorem proving, it may be beneficial to include shorter proofs whenever possible, since this should improve performance.
> > >
> > > **Q3.**
> > >
> > > Formal proofs are represented by finite words. The set of words of length bounded by a fixed finite constant is compact. In addition, actual datasets are finite, and therefore compact as well.
> > >
> > > More importantly, compactness is used as a **sufficient condition** to guarantee the existence of an optimal policy. Note that non-compactness does **not** imply that an optimal policy does not exist; compactness is simply a convenient and mild sufficient assumption for our analysis.
> > >
> > > **Q6.**
> > >
> > > In typical theorem-proving settings, $\mathcal X \times \mathcal A$ is a set of words, hence a discrete set.
> > >
> > > Any map on a discrete set is automatically continuous. Therefore, unless one intentionally introduces a discontinuous embedding, the continuity condition is naturally satisfied.
> > >
> > > **Q7/Q8.**
> > >
> > > > *This explanation needs a lot of work.*
> > >
> > > We respectfully disagree that the point is purely vague.
> > >
> > > In this paper, we introduce the notion of **proof success probability** in a simplified setting for agentic provers, and we analyze it theoretically. More complex settings would generally be expected to improve upon this baseline, not invalidate it. In that sense, our result should be viewed as a baseline theory.
> > >
> > > We believe that this is already a substantial contribution for a single conference paper.

---

### Official Review · Reviewer_chk9 · 2026-03-12

**Soundness:** 3
**Presentation:** 2
**Significance:** 3
**Originality:** 3
**Overall Recommendation:** 5
**Confidence:** 3

**Summary:**

Main contributions: The paper is a theoretical work on formalizing agentic theorem provers as time-bounded reachability Markov decision processes (MDPs). The authors give a variety of results related to this formalism, proving, for instance, lower and upper bounds for the probability of the provability of a problem. This framework covers many different parts relating to theorem proving pipelines, such as retrieval over libraries, test-time search and representation geometry.

**Compliance With Llm Reviewing Policy:**

Affirmed.

**Final Justification:**

The paper is interesting and the topic is timely. The authors' approach is innovative. The principled rebuttal resolved the concerns that I had.

**Key Questions For Authors:**

Where is the proof of Theorem 9.1? What is the main message of the paper from the technical point of view, i.e., which theorem(s) would you consider your main result? What future work would this paper support?

**Limitations:**

An explicit discussion of limitations would improve the work.

**Strengths And Weaknesses:**

The paper is original and concerns a timely and important topic. The change in perspective (focusing on real-life complexity of "typical" cases instead of worst-case complexity) is refreshing and brings the theoretical work closer to what happens in actual practice.

The paper is substantial and contains a wide range of results.

Concerning potential weaknesses, the paper is mathematically very dense. Producing a conceptual narrative in addition to just listing results would improve readability. The section numbering is a little bit dense as well, with more sections than pages, and subsections sometimes consist of a single sentence. There are also typos and typesetting flaws (e.g. the overflowing equation on page 8), but these are minor issues.

To improve the presentation, perhaps reworking the core structure into 5-6 key sections could help. Also, more intuition, with more of the mathematical jargon placed in the appendix, would help the reader. Currently, much of the intuition and expository content is placed in the Appendix, along with the literature review, and as a result, the main paper is can be difficult to understand by itself.

---

> ### Author Rebuttal · Authors · 2026-03-28
>
> Thank you for the positive assessment and for the very helpful suggestions on presentation. We are glad that you found the topic timely and that the shift from worst-case complexity to a “typical-case” / distributional viewpoint seems valuable. We also agree with your main presentation concern: the current manuscript is too dense. In the revised manuscript, we will streamline the section structure, add more intuition in the main text, and make the theorem-to-message map much more explicit.
>
> Regarding Theorem 9.1, we apologize that the appendix/proof pointer was omitted in the submitted version. In the revised manuscript, we will state the proof location explicitly and include a short proof sketch in the main text. At a high level, the argument has three ingredients. First, we prove an adaptive uniform deviation lemma for non-i.i.d. data collected by exploration: after taking an $\varepsilon$-net of the function class in sup norm, we apply Azuma-Hoeffding to the resulting bounded martingale differences and then take a union bound over the net. Second, we bound the metric entropy of the Lipschitz score class on a domain $D$, which controls the size of the $\varepsilon$-net in terms of $d_D$ and $L_H$. Third, we combine this deviation bound with the approximation term through an ERM/oracle-inequality argument to obtain
>
> \\[
> \sup_{(\mu,a)\in D} |\hat h_b(\mu,a)-Q_b^*(\mu,a)| \le \epsilon_{\mathrm{app},b} + \epsilon_{\mathrm{stat}}(n,\delta)
> \\]
>
> up to logarithmic factors. We will also clarify how the targets $Y$ arise in theorem proving: in the idealized statement, they can be defined as bounded rollout-success labels whose conditional expectation is $Q_b^\ast$; practically, repeated rollouts or other value-target constructions can be used to approximate them.
>
> From the technical point of view, the main contribution of this paper is not a claim that one specific search heuristic is universally best. Rather, we intentionally simplify agentic proving in order to exhibit a new framework: **statistical provability**, i.e., time-bounded goal-reaching probability for theorem-proving agents. If we had to identify the technical core, it is the chain of results showing that: (i) Bellman structure gives provability certificates; (ii) score-guided planning methods such as greedy / top-$k$ / beam admit deviation bounds in terms of score error and margin; and (iii) these score errors themselves admit high-probability control under geometric / statistical assumptions. In that sense, Theorem 7.1, Theorems 8.1--8.2, and Theorem 9.1 together form the main technical message. We agree that this is not stated clearly enough in the current version, and we will make it explicit in the revised manuscript.
>
> As for future work, the present paper is meant as a base layer rather than a full theory of all prover systems. As discussed in Appendix A.3, practical agentic theorem provers use many additional mechanisms and heuristics beyond the simplified abstraction analyzed here. A natural next step is therefore to instantiate the framework under stronger assumptions on the search algorithm, the learning algorithm, and the theorem distribution. This would make it possible to estimate statistical provability more directly from prover traces and to use it as a diagnostic for improving the agent itself---for example, by identifying whether the main bottleneck is effective horizon, approximation error, data coverage, weak action margins, or search instability. We will add this motivation and future-work direction more clearly in the revised manuscript.

---

> > ### Author Rebuttal · Reviewer_chk9 · 2026-04-03
> >
> > Thank you for the rebuttal. The concern with Theorem 9.1 was fully resolved, and the key technical contributions clearly listed and also elucidated. Moreover, I appreciate that the authors better explained the position of this paper in relation to their longer-term aims.

---

### Official Review · Reviewer_jZGT · 2026-03-13

**Soundness:** 2
**Presentation:** 1
**Significance:** 3
**Originality:** 2
**Overall Recommendation:** 2
**Confidence:** 3

**Summary:**

This work considers a major problem: explaining the empirical success of LLM-based theorem provers while complexity theory suggests proof search should be intractable. They introduced a statistical provability framework, where theorem proving is modeled as a finite-horizon reachability MDP whose objective is the probability of completing a proof within a compute budget. They define provability as the optimal success probability of reaching a solved proof state and analyze it using Bellman equations.

The paper proves the existence of optimal policies under mild conditions, derives upper/lower bounds on success probability, and provides theoretical bounds showing how approximation error, representation geometry, search strategies, retrieval, and proof decomposition affect the probability of success.

**Compliance With Llm Reviewing Policy:**

Affirmed.

**Key Questions For Authors:**

### Questions

1. Could the authors explain the logical flow from one theorem to the next, especially how the results in Sections 5–9 lead to the conclusions in Section 11? If the conclusions are indeed supported by the theory, then the analysis is quite aligned with intuition and could be useful for guiding or analyzing future work. However, in the current presentation, that connection is not clear.

2. Could the authors clarify the novelty and significance of their work relative to the following related papers?

   [1] Matthieu Zimmer et al. Bourbaki: Self-Generated and Goal-Conditioned MDPs for Theorem Proving.

   [2] Sho Sonoda et al. Don’t Eliminate Cut: Exponential Separations in LLM-Based Theorem Proving.

**Limitations:**

Yes

**Strengths And Weaknesses:**

### Strengths

1. Well-motivated, because the authors consider a major problem: explaining why modern agentic theorem provers (LLM + search + verification) work well in practice
2. The finite-horizon reachability MDP was reasonable. The author tried to define the notion of statistical provability, as the probability of completing a proof within a finite compute budget averaged over an instance distribution, shifting from worst-case complexity toward distributional success.

---

### Weaknesses

The paper is overall poorly written. In particular, Sections 5–9 consist almost entirely of definitions, lemmas, theorems, and proofs, with very little bridging text or intuitive explanation. This makes the paper difficult to follow, and as a result, it is also hard to evaluate whether the technical development is correct and whether the assumptions are reasonable.


The paper models theorem proving as a finite-horizon MDP. While this is a clean abstraction, the idea of viewing theorem proving as an MDP is not new. It has already appeared in several pre-LLM works on learning-guided theorem proving, as well as in more recent work such as [1]. They seem to weaken the novelty of the paper.

[1] Matthieu Zimmer et al. Bourbaki: Self-Generated and Goal-Conditioned MDPs for Theorem Proving.

---

> ### Author Rebuttal · Authors · 2026-03-28
>
> Thank you for the thoughtful and constructive feedback. We agree that the current presentation does not make the logical flow from Sections 5–9 to Section 11 sufficiently clear, and this is something we will address in the revised manuscript.
>
> > 1. Could the authors explain the logical flow ...
>
> The intended flow is the following. Section 5 defines the main object of the paper: time-bounded provability as a finite-horizon reachability value function $V_B^\ast$. Section 6 then shows that, under mild compactness/continuity assumptions, the Bellman maximization is attained, so the quantity defined in Section 5 is realized by optimal deterministic Markov policies rather than remaining only a formal supremum. Section 7 uses this Bellman structure to derive upper/lower provability certificates, i.e., quantities that sandwich $V_B^\ast$ without solving the full MDP. Section 8 then connects these certificates to score-guided planning: if the score $h_b$ approximates the optimal one-step value $Q_b^\ast$ uniformly on a relevant domain, then greedy / top-$k$ / beam-style search has controlled deviation from optimal provability, with stronger fast-rate behavior under margin conditions. Section 9 provides sufficient conditions under which the uniform approximation premise in Section 8 can hold with high probability, by decomposing the error into approximation and statistical terms governed by the geometry of the relevant domain and the hypothesis class. Section 11 is therefore intended as an interpretation of the parameters appearing in Sections 8–9: effective horizon/proof length, approximation error, statistical complexity, margins, and coverage / overflow terms. In other words, the “easy vs. hard instances” discussion is not meant as a separate theorem, but as a qualitative reading of the formal upper/lower bounds developed earlier. We agree that this theorem-to-conclusion map should be stated much more explicitly, and in the revised manuscript we will add a roadmap paragraph and more bridging text to make this dependency transparent.
>
> > 2. Could the authors clarify the novelty and significance of their work relative to the following related papers?
>
> We also agree that the MDP viewpoint itself is not the main novelty, and we will revise the related-work discussion accordingly.
>
> Relative to [1], our work differs in both goal and methodology. Bourbaki also formulates theorem proving in an MDP-style framework, but its main contribution is an experimental/search framework based on self-generated goal-conditioned MDPs together with MCTS-like search. By contrast, our paper is a theory paper: the focus is not on implementing a stronger search system, but on analyzing time-bounded provability and the effect of score-guided search procedures such as top-$k$ through Bellman inequalities, certificate bounds, and approximation/statistical error terms.
>
> Relative to [2], there is again overlap in the broad MDP formalization and in the motivation that structural proof organization matters. However, [2] focuses on a hierarchical cut-aware vs. flat cut-free comparison and proves an exponential separation in data/sample requirements in that setting. Our paper instead develops upper/lower provability bounds for score-guided planning and derives fast-rate-type guarantees under margin conditions via Bellman certificates and uniform approximation arguments. 　We will add both papers to the related-work discussion and make these distinctions explicit in the revised manuscript.

---

### Decision · Program_Chairs · 2026-04-30

**Decision:**

Accept (regular)

**Comment:**

Reviewers agreed that this paper tackles an important problem: providing a foundational theory explaining the success of agentic theorem provers. To do this, the paper introduces the idea of statistical provability, the probability of success in proving an instance under a compute budget averaged over an instance distribution, and shows several results which allow to reason about provability without explicitly compute the agent behavior. The paper provides an interesting solid theory for explaining agentic theorem provers, and the paper present strong technical results. The presentation needs improvements but overall the results of this paper are a strong contributions for the ICML community.